# Correcting atmospheric $CO_2$ and $CH_4$ mole fractions obtained with Picarro analyzers for sensitivity of cavity pressure to water vapor

Friedemann Reum[1], Christoph Gerbig[1], Jost V. Lavric[1], Chris W. Rella[2] and Mathias Göckede[1]

[1]Max Planck Institute for Biogeochemistry, Jena, Germany
[2]Picarro Inc., Santa Clara, CA, USA

*Correspondence to*: Friedemann Reum (freum@bgc-jena.mpg.de)

**Abstract.** Measurements of dry air mole fractions of atmospheric greenhouse gases are used in inverse models of atmospheric tracer transport to quantify their sources and sinks. The measurements have to be calibrated to a common scale to avoid bias in the inferred fluxes. For this purpose, the World Meteorological Organization (WMO) has set requirements

for the inter-laboratory compatibility of atmospheric greenhouse gas (GHG) measurements. A widely used series of devices for these measurements are the GHG analyzers manufactured by Picarro, Inc. These are often operated in humid air, and the effects of water vapor are corrected for in post-processing. Here, we report on rarely detected and previously unexplained biases of the water correction method for $CO_2$ and $CH_4$ in the literature. They are largest at water vapor mole fractions below 0.5 % $H_2O$, which were undersampled in previous studies, and can therefore affect measurements obtained in humid air.

Setups that dry sample air using Nafion membranes may be affected as well if there are differences in residual water vapor levels between sample and calibration air. The biases are caused by a sensitivity of the pressure in the measurement cavity to water vapor. We correct these biases by modifying the water correction method from the literature. Our method relies on experiments that maintain stable water vapor levels to allow equilibration of cavity pressure. In our experiments with the commonly used droplet method, this requirement was not fulfilled. Correcting $CO_2$ measurements proved challenging,

presumably because of our humidification method. Open questions pertain to differences between analyzers and variability over time. In our experiments, the biases amounted to considerable fractions of the WMO inter-laboratory compatibility goals. Since measurements of dry air mole fractions of $CO_2$ and $CH_4$ are also subject to other uncertainties, correcting the cavity pressure-related biases helps keeping the overall accuracy of measurements obtained with Picarro GHG analyzers in humid and potentially in Nafion-dried air within the WMO goals.

## 1    Introduction

Measurements of atmospheric GHG mole fractions are integral data for quantifying their sources and sinks using inverse models of atmospheric transport (e.g. Kirschke et al., 2013; McGuire et al., 2012). Inverse models require atmospheric measurements calibrated to a common scale, because relative biases in the atmospheric mole fractions lead to biases in the inferred fluxes. To ensure the high quality of greenhouse gas observations required for inverse models of atmospheric

transport, the World Meteorological Organization (WMO) has set compatibility goals for atmospheric $CO_2$ and $CH_4$ measurements to ±0.1 ppm for $CO_2$ (±0.05 ppm in the southern hemisphere) and ±2 ppb for $CH_4$ (WMO, 2016) between laboratories. This compatibility is ensured if individual laboratories keep errors of measurements with respect to a common calibration scale below half of these goals, which corresponds to the so-called internal reproducibility goals (WMO, 2016).

Models of atmospheric greenhouse gas transport require dry air mole fractions as input, i.e. the number of molecules of the target gas divided by the number of air molecules excluding water vapor. Water vapor is excluded because its variability would mask signals in the greenhouse gases.

GHG analyzers manufactured by Picarro Inc. (Santa Clara, CA), which are based on the cavity ring-down spectroscopy technique (Crosson, 2008), are used at many GHG monitoring sites because of their signal stability. Due to limitations of air

sample drying techniques (Rella et al., 2013), these analyzers are often operated in humid air, and dry air mole fractions are obtained by correcting for the effects of water vapor in a post-processing step (Chen et al., 2010; Rella et al., 2013). The effect of water vapor on trace gas readings can be described by a water correction function $f_c(h)$, where $c$ denotes the target gas (here: $CO_2$ or $CH_4$) and $h$ is the water vapor mole fraction (measured by the Picarro analyzer). The analyzer reports wet air mole fractions $c_{wet}(h)$, from which dry air mole fractions $c_{dry}$ can be obtained by dividing by the water correction

function:

$$c_{dry} = \frac{c_{wet}(h)}{f_c(h)} \tag{1}$$

The water correction function from the literature takes into account dilution and line shape effects. These are described by a second-degree Taylor series, i.e. a parabola (Chen et al., 2010; Rella et al., 2013):

$$f_c^{para}(h) = 1 + a_c \cdot h + b_c \cdot h^2 \tag{2}$$

Thus, dry air mole fractions based on this model are calculated as:

$$c_{dry}^{standard} = \frac{c_{wet}(h)}{f_c^{para}(h)} \tag{3}$$

Henceforth, we call this the "standard" water correction model.

In previous studies featuring water corrections for $CO_2$ and $CH_4$, water vapor mole fractions below 0.5 % $H_2O$ were only scarcely sampled (Chen et al., 2010; Nara et al., 2012; Rella et al., 2013; Winderlich et al., 2010). In this paper, we report on biases in $c_{dry}^{standard}$ in this domain that were not detected in these previous studies. They were, however, recently detected in one other study in which this domain was sufficiently sampled (Stavert et al., 2018). We hypothesize that the biases in $CO_2$ and $CH_4$ readings are due to an as yet undocumented sensitivity of the pressure inside the measurement cavity to water

vapor. We designed and conducted experiments that uncovered that the internal pressure sensor, which is used to stabilize cavity pressure, produces erroneous readings in the presence of water vapor. These errors cause a sensitivity of cavity pressure to water vapor that translates into biases in $CO_2$ and $CH_4$ readings. Thus, the hypothesis was confirmed. Based on these results, we provide an approach to correct the biases in $CO_2$ and $CH_4$ readings. We also discuss remaining challenges,

which are related to the reliable correction of $CO_2$ readings as well as differences between analyzers and variability over time.

## 2 Materials and Methods

To determine the effect of water vapor on $CO_2$ and $CH_4$ measurements obtained using Picarro analyzers, as well as on the pressure in the measurement cavity, so-called "water correction" experiments similar to those in the literature (e.g. Rella et al., 2013) were performed, i.e. dry air from pressurized gas tanks was humidified and measured with Picarro GHG analyzers. Dry air mole fractions used were in the ranges 352–426 ppm $CO_2$ and 1797–2115 ppb $CH_4$. The key modifications to the experiments in the literature were to monitor cavity pressure independently of the internally mounted cavity pressure sensor in some experiments and more densely sample at water vapor mole fractions below 0.5 % $H_2O$. Experiments were performed with five Picarro GHG analyzers, henceforth labeled "Picarro #1 – #5", and one Picarro oxygen analyzer labeled "Picarro #6" (Table 1). The setup varied between experiments (Table 1, Fig. 1–Fig. 3) because of analyzer type (see Sect. 2.1 for a brief explanation) and because experiments were performed at different stages of this study with different goals (see caption of Table 1). In the following sections, we first describe relevant aspects of the measurement principle and hardware of Picarro analyzers, and then describe our experiments.

Table 1: Overview of experiments performed for this study. Experiments with Picarros #1 and #2 were conducted at an early stage of this work and were designed to solely characterize the cavity pressure dependence on water vapor. Therefore, the experiments with stable $H_2O$ levels with these analyzers did not yield trace gas readings suitable for analysis (column 5). Experiments with Picarros #4 and #5 were performed without independent pressure monitoring for reasons stated below. Spectroscopic cavity pressure measurements were not possible with Picarro GHG analyzers (see Sect. 2.3.2).

| Label | Picarro analyzer model | Picarro analyzer type | Droplet experiment with external pressure measurement | Stable $H_2O$ level experiment: external cavity pressure measurement / usable trace gas measurements (reason) | Stable $H_2O$ level experiment: spectroscopic cavity pressure measurements |
|---|---|---|---|---|---|
| #1 | G2401-m | Flight-ready | Yes | Yes / No (used ambient air) | No |
| #2 | G2401 | Regular | No | Yes / No (disregarded equilibration) | No |
| #3 | G2401-m | Flight-ready | No | Yes / Yes | No |
| #4 | G2401-m | Flight-ready | No | No (conducted before cavity pressure hypothesis was developed) / Yes | No |
| #5 | G2301 | Regular | No | No (remote field site) / Yes | No |
| #6 | G2207-i | Regular | No | No (replaced by spectroscopic measurements) / No (analyzer measures oxygen, not $CO_2$ and $CH_4$) | Yes |

## 2.1 Picarro GHG analyzers: measurement principle and active cavity pressure stabilization system

Picarro GHG analyzers are based on the cavity ring-down spectroscopy method (Crosson, 2008). In a measurement cavity, laser pulses scan absorption lines of the target gases. The time it takes the pulses to attenuate is converted to mole fractions of the gases. Among other requirements, the analysis assumes stable pressure inside the measurement cavity. Cavity pressure stability is achieved by a feedback loop (e.g. Fig. 1) between a pressure sensor (General Electric NPC-1210) that is mounted inside the cavity, and the outlet valve of the cavity (inlet valve in so-called flight-ready Picarro GHG analyzers, which are customized for airborne measurements). This loop keeps readings of the cavity pressure sensor stable. Picarro GHG analyzers for $CO_2$ and $CH_4$ used in this study, i.e. model series G2301 and G2401, operate at 186.65 hPa (140 Torr) with a $1\sigma$ tolerance of 0.20 hPa.

## 2.2 Setups for humidification

To humidify the air stream, two different methods were used. The first approach was designed to maintain stable water vapor levels, while the second approach was the commonly used droplet method. In this section, we describe the experimental setup for both methods.

### 2.2.1 Stable water vapor levels

To create an air stream with stable water vapor levels, the dry air stream was split into two lines, one of which remained untreated. Air in the other line was directed through a gas washing bottle that contained deionized water (e.g. Fig. 1). For experiments where $CO_2$ and $CH_4$ data were analyzed, the amount of water used was 15 ml (Picarro #3) or 40 ml (Picarros #4 and #5). With this method, air in the humidified line was saturated with water vapor (mole fraction ~3 % $H_2O$). Subsequently, the two lines were joined again. The water vapor mole fraction in the re-joined line was controlled by adjusting the flow through the wet and dry lines. In the experiments with Picarros #1–#5, this was achieved using needle valves; in the experiment with Picarro #6, mass flow controllers (Alicat Scientific, Tucson, Arizona) were used. In an experiment with Picarro #1 that was conducted at an early stage of this work, instead of using the gas washing bottle approach, stable water vapor levels were realized by mixing air from the gas tank with ambient laboratory air. The experiment solely served to characterize the cavity pressure dependence on water vapor; $CO_2$ and $CH_4$ readings from this experiment were not analyzed.

### 2.2.2 Droplet method

For droplet experiments, the humidification unit described above was replaced with a tee piece that enabled injecting water droplets into the dry air stream (Fig. 2).

## 2.3    Setups for cavity pressure monitoring

We used two methods to monitor pressure inside the measurement cavity independently of the internally mounted pressure sensor. The first method was based on an additional pressure sensor. Due to the complexity of this setup, we developed a second cavity pressure monitoring method, based on spectroscopic measurements, to verify the results of the first approach. In this section, we describe the experimental setups for both methods.

### 2.3.1    Cavity pressure monitoring with external sensor

For this approach, cavity pressure was monitored with an additional pressure sensor (General Electric Druck DPI 142). The optimal placement of this sensor would be between cavity and inlet or outlet valve, as this position would expose it directly to cavity pressure changes. However, opening tubing connections at these positions would risk contaminating the cavity, which would be expensive and time-consuming to fix. In addition, this setup could interfere with temperature control of the cavity by introducing a heat bridge and may thus require modifying the Picarro analyzer. For these reasons, the external pressure sensor was installed outside of the Picarro analyzer (e.g. Fig. 1). To ensure that the external sensor could react to changes in cavity pressure, it was installed adjacent to the cavity valve that was not used to control cavity pressure, i.e. upstream of the inlet valve in experiments with "regular" analyzers (Fig. 1) and downstream of the outlet valve in experiments with "flight-ready" analyzers (Fig. 2). During normal operation, the inlet and outlet valves act as chokes and would thus shield the external pressure sensor from cavity pressure changes. Therefore, pressure in the external pressure measurement branch was adjusted to within a few hPa of cavity pressure by installing a needle valve as a choke (e.g. Fig. 1). This way, the valve between cavity and external pressure sensor did not act as a choke and the sensor could react to cavity pressure changes. Since the external pressure sensor may itself be sensitive to water vapor, it was shielded from humidity changes by installing it behind a drying cartridge filled with magnesium perchlorate in a dead end (e.g. Fig. 1). This setup allowed monitoring cavity pressure independently of water vapor content, while the internal cavity pressure sensor still reacted to changes in water vapor levels in the sampling air. The relationship between readings of the external pressure sensor and cavity pressure changes was calibrated in separate experiments with constant humidity (Sect. 2.4).

### 2.3.2    Cavity pressure monitoring with spectroscopic methods

Cavity pressure of Picarro analyzers affects the width of absorption lines used to measure target gas mole fractions, and the optical phase length (physical path length times refractive index) of the measurement cavity. Both quantities were used to monitor cavity pressure.

The $CO_2$ absorption line is not a good choice for this experiment, because it has a strong line broadening effect with water vapor (Chen et al., 2010). The $CH_4$ absorption feature is also a poor choice, because it is not a clean, isolated line. Instead, a CRDS analyzer measuring $O_2$, $\delta^{18}O$ and $H_2O$ (G2207-i, Picarro, Inc., Santa Clara), which works with an $O_2$ absorption line at 7878.805547 cm$^{-1}$ (John Hoffnagle, personal communication), was used. The active cavity pressure stabilization system of

this analyzer is identical to that of Picarro GHG analyzers with the exception that it operates at 339.97 hPa (255 Torr) rather than 186.65 hPa. Therefore, we expect the dependence of cavity pressure on water vapor of this analyzer to be of similar magnitude and form as for GHG analyzers.

Both $O_2$ line width and optical phase length are also influenced directly by water vapor: pressure broadening of absorption line widths has been shown in a variety of systems to be linearly dependent upon the background gas matrix, and in particular on water vapor (Chen et al., 2010; Johnson and Rella, 2017; Nara et al., 2012). We therefore expect a linear dependence of the $O_2$ line width on water vapor mole fraction. Similarly, the index of refraction of air also depends on the gas matrix (Chen et al., 2016), leading to a linear dependence of the optical phase length on water vapor mole fraction. Hence, we attribute non-linear dependencies of $O_2$ line width and optical phase length on water vapor to changes in cavity pressure.

### 2.4    Experiments for inferring sensitivities to varying cavity pressure

To determine how readings of the external pressure sensor, $CO_2$, $CH_4$, and $H_2O$ of the Picarro GHG analyzers, and $O_2$ line width and optical phase length of the oxygen analyzer react to changes in internal cavity pressure, calibration experiments were performed. For these experiments, air from a gas tank was measured with the Picarro analyzer. Initial equilibration periods of readings from the external pressure sensor, $CO_2$ and $CH_4$ (GHG analyzers), and of $O_2$ line width and optical phase length (oxygen analyzer) were discarded. Then, cavity pressure was varied using Picarro Inc. software. Cavity pressure levels were chosen so that the range spanned between dry and humid air as retrieved with the external pressure sensor in water correction experiments was covered, and probed for several minutes each. Most sensitivity experiments were performed with dry air. With Picarro #3, an additional sensitivity experiment was performed at a water vapor level of 3 % $H_2O$. With Picarros #4 and #5, no sensitivity tests were performed because no experiments with external pressure monitoring were performed with these analyzers. This was because the experiments with Picarro #4 were performed before the cavity pressure hypothesis was developed, and Picarro #5 was operated at a remote field site.

### 2.5    Water correction experiments with external pressure monitoring

**Experiments with stable water vapor levels**

During stable water vapor level experiments with external pressure monitoring, water vapor levels were probed between 15 and 150 minutes (median about 40 minutes) depending on the stability of the external pressure measurement and trace gas readings. External pressure readings drifted on a timescale of several hours relative to internal cavity pressure readings. Therefore, external pressure sensor readings obtained in humid air were calibrated against external pressure sensor readings in dry air by probing dry air before and after each measurement in humid air. For further analysis, average readings from the Picarro GHG analyzer and the external pressure sensor of the last 10 minutes of each probing interval were used to reduce noise (15 minutes during the experiment with Picarro #3, five minutes for some low water vapor levels with Picarro #1). The order of water vapor levels was altered between experiments, including high–low–high patterns and random alternations.

Varying water levels monotonically throughout an experiment was avoided to ensure that the influence of various potential error sources was not systematic (Sect. S3).

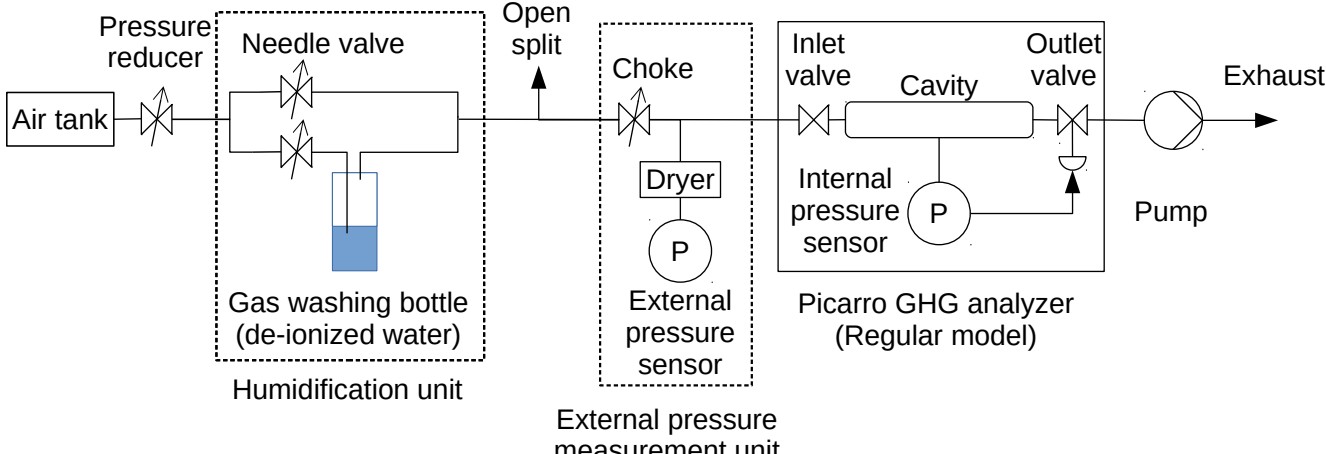

**Fig. 1: Experimental setup for experiments with stable water vapor levels and external pressure monitoring. Shown here is the setup for a regular Picarro GHG analyzer (Picarro #2), from which only pressure data were analyzed. For Flight-ready analyzers, the external pressure measurement unit was placed downstream of the analyzer (Fig. 2).**

**Droplet experiments**

Droplet experiments with external pressure monitoring were performed with Picarro #1 using the setup shown in Fig. 2. For each droplet experiment, the tee piece was opened, a droplet of deionized water (~ 1 ml) was injected using a syringe, and the tee piece was closed. Gradual evaporation of this water droplet then caused a gradient over time from high to low water vapor levels in the sample air.

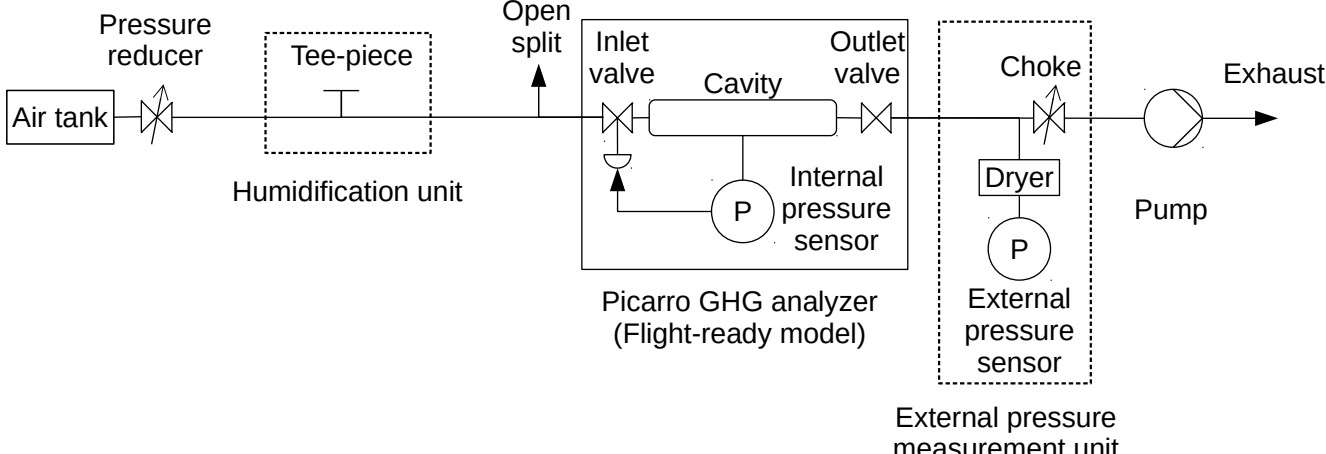

Fig. 2: Experimental setup for water correction experiments with humidification via water droplets and external pressure monitoring. Here, the setup for a flight-ready analyzer is shown.

5  ## 2.6    Experiments for spectroscopic cavity pressure measurements

For spectroscopic cavity pressure measurements, water vapor was ramped up and down with a period of about 240 minutes for several cycles using the setup depicted in Fig. 3. Two ranges of water vapor mole fractions were selected for the experiment: a narrow range (0–0.2 % $H_2O$) for sampling the pressure bend at high resolution for five cycles, and a wider range up to about 0.8 % $H_2O$ for another six cycles to establish the transition to a linear dependence of the pressure proxies

10  $O_2$ line width and optical phase length on water vapor mole fraction.

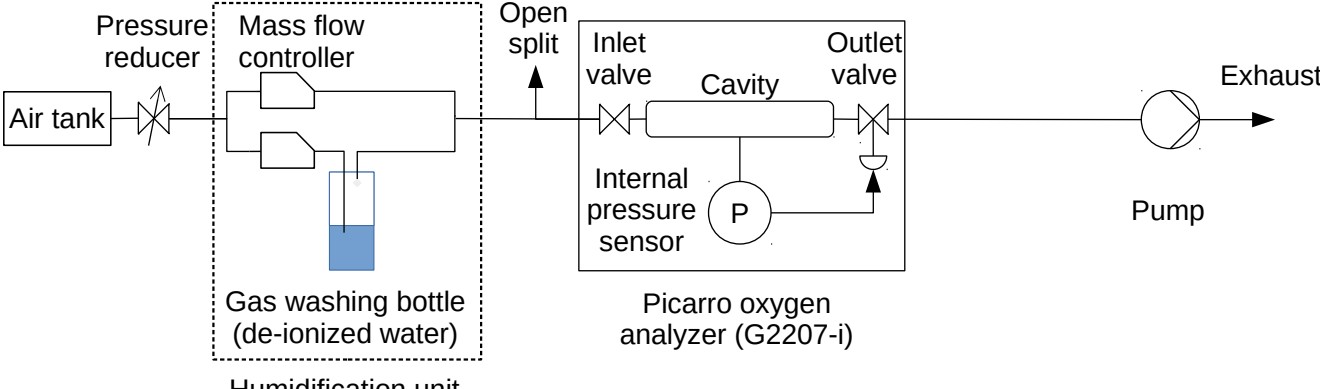

**Fig. 3: Experimental setup for spectroscopic cavity pressure measurements.**

## 3    Results

In this section, we first demonstrate the relevance of cavity pressure for $CO_2$ and $CH_4$ measurements performed with Picarro GHG analyzers and establish the sensitivities of the independent pressure monitoring methods to changes in cavity pressure (Sect. 3.1). We then present our results on the dependency of cavity pressure on water vapor (Sect. 3.2), and introduce modifications to the standard water correction model for $CO_2$ and $CH_4$ that account for this sensitivity (Sect. 3.3). Finally, we examine the performance of standard and modified water correction models in water correction experiments with stable water vapor levels (Sect. 3.4) and droplet experiments (Sect. 3.5).

### 3.1    Sensitivities of independent pressure measurements and trace gas readings to changes of internal cavity pressure

In the sensitivity tests with Picarro GHG analyzers, readings from the external pressure sensor, as well as of $CO_2$ and $CH_4$ all varied linearly with cavity pressure, demonstrating that biases in cavity pressure directly affect mole fraction readings. Similar sensitivities were observed for all analyzers (Table 2). On average, for dry air mole fractions of 400 ppm $CO_2$ and 2000 ppb $CH_4$, a change of 1 hPa in cavity pressure would cause a difference of 0.37 ppm $CO_2$ and 6.4 ppb $CH_4$. The sensitivities obtained in the experiment with humid air (3 % $H_2O$) differed by only a few percent from those obtained in dry air with the same analyzer ($CO_2$: +5 %, $CH_4$: -2 %, external pressure readings: -1 %). Hence, all sensitivities were treated as independent of the water vapor mole fraction.

In the sensitivity tests with the oxygen analyzer, both the $O_2$ line width and the optical phase length of the cavity varied linearly with cavity pressure, with the sensitivities shown in Table 2.

**Table 2: Sensitivities of readings of Picarro GHG analyzers and independent pressure measurements to variations of internal cavity pressure $p$. For the quantities pertaining to GHG analyzers, averages and standard deviations of all sensitivity experiments are reported, while for the quantities pertaining to the $O_2$ analyzer, mean and standard error of the fit of the single experiment are given.**

| Quantity | Analyzer | Sensitivity to cavity pressure |
| --- | --- | --- |
| External pressure measurement $\left(\frac{\partial p_{ext}}{\partial p}\right)$ | #1–#3 | $(0.95 \pm 0.04)$ hPa hPa$^{-1}$ |
| $CO_2 \left(\frac{\partial CO_2}{\partial p}/CO_2^{dry}\right)$ | #1–#3 | $(9.2 \pm 0.3) \times 10^{-4}$ hPa$^{-1}$ |
| $CH_4 \left(\frac{\partial CH_4}{\partial p}/CH_4^{dry}\right)$ | #1–#3 | $(3.22 \pm 0.05) \times 10^{-3}$ hPa$^{-1}$ |
| $O_2$ line width | #6 | $(4.05 \pm 0.05) \times 10^{-3}$ hPa$^{-1}$ |
| Optical phase length | #6 | $(163 \pm 3)$ nm hPa$^{-1}$ |

## 3.2 Dependency of cavity pressure on water vapor

### 3.2.1 Results from external pressure sensor (stable water vapor levels)

**Experimental results**

Cavity pressure was monitored with the external sensor during experiments with stable water vapor levels with three different Picarro GHG analyzers. Readings of the internally mounted cavity pressure sensors were, owing to the active pressure stabilization system of the analyzers, stable at 186.65 hPa with standard deviations of 0.02 hPa or less (as expected). However, cavity pressure as estimated based on external pressure sensor readings and their sensitivity to cavity pressure variations (Sect. 3.1) varied systematically with the water vapor mole fraction, revealing that the readings of the internal sensors were biased in the presence of water vapor. Cavity pressure estimated based on the external sensor displayed a uniform pattern for all three analyzers (Fig. 4): cavity pressure decreased when the water vapor level increased, and the gradient of the variation was larger below about 0.2 % $H_2O$, which created a bend in the dependency of cavity pressure on water vapor (henceforth called "pressure bend").

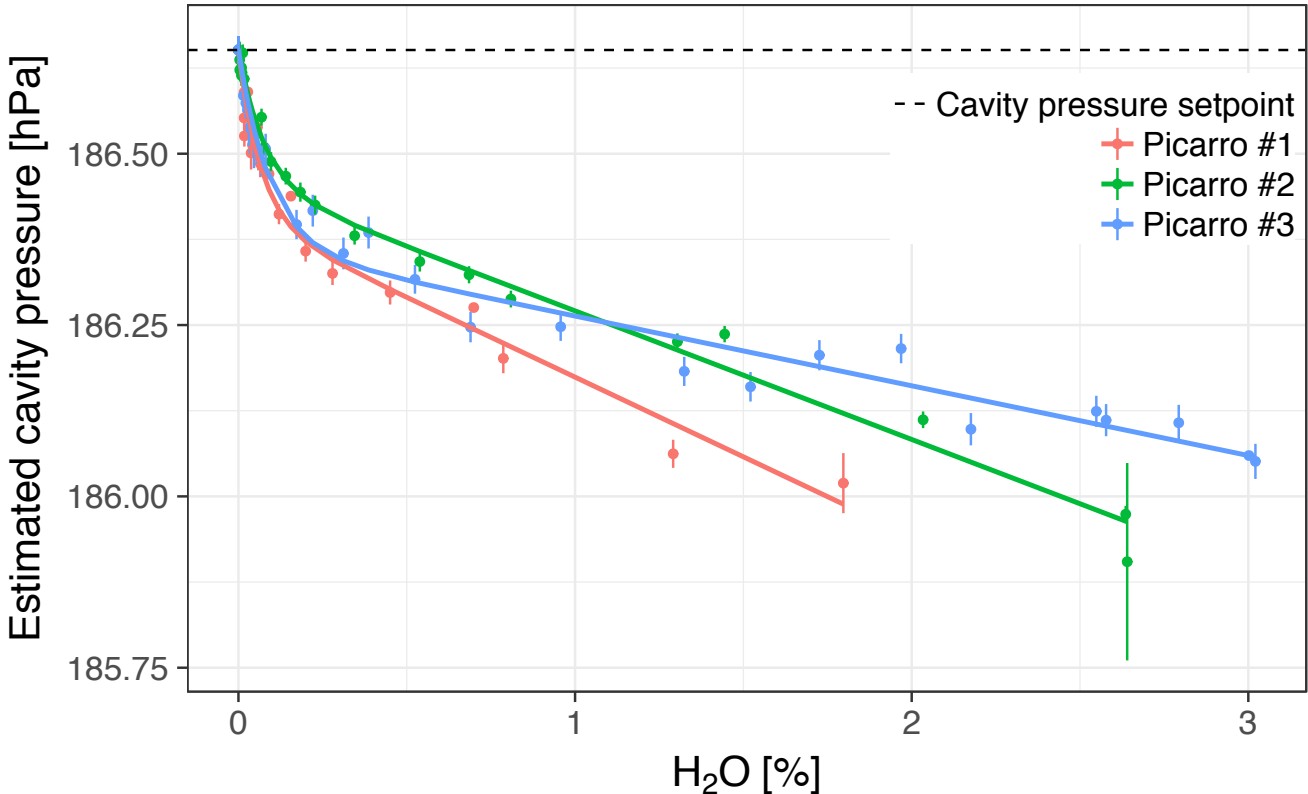

**Fig. 4: Cavity pressure estimated based on external pressure sensor readings in experiments with stable water vapor levels and fits of the empirical cavity pressure model Eq. (4) to the data. Error bars: lower bound of uncertainty; see Sect. S1.2.**

**Empirical description**

Based on these results, we formulated an empirical description of cavity pressure dependency on water vapor:

$$p_{est}(h) = p_0 + s \cdot h + d_p \cdot \left( e^{-\frac{h}{h_p}} - 1 \right) \tag{4}$$

In this equation, $p_{est}$ is the estimated cavity pressure, $h$ is the water vapor mole fraction, $p_0$ is the cavity pressure in dry air (186.65 hPa for Picarro GHG analyzers), $h_p$ is the position of the pressure bend, $s$ is the slope for $h \gg h_p$, and $d_p$ describes the magnitude of the pressure bend.

The empirical cavity pressure model Eq. (4) was fitted to the data of each analyzer. The coefficient of determination was larger than 0.98 for all experiments, indicating good fits. Estimated coefficients varied between analyzers (Table 3).

**Table 3: Coefficients of the empirical cavity pressure model Eq. (4) for data from experiments with stable water vapor levels and external pressure monitoring (estimate and standard error). The last line shows averages and standard deviations of the individual estimates.**

| Analyzer | $s$ [hPa (% $H_2O$)$^{-1}$] | $h_p$ [% $H_2O$] | $d_p$ [hPa] |
|----------|------------------------------|-------------------|--------------|
| #1 | -0.131 ± 0.009 | 0.066 ± 0.009 | 0.245 ± 0.016 |
| #2 | -0.106 ± 0.003 | 0.076 ± 0.009 | 0.193 ± 0.009 |
| #3 | -0.057 ± 0.004 | 0.095 ± 0.011 | 0.286 ± 0.012 |
| Average | -0.10 ± 0.04 | 0.079 ± 0.014 | 0.24 ± 0.05 |

### 3.2.2 Results from external pressure sensor during droplet experiments

Cavity pressure estimated based on external pressure sensor readings varied strongly between droplet experiments and was consistently lower than during the stable water vapor level experiment with this analyzer (Fig. 5, top panel). The largest variations occurred below 1 % $H_2O$. In this domain, the droplets dried up quickly, which caused very fast decreases of the water vapor mole fraction from about 0.5–1 % to 0 % $H_2O$ (Fig. 5, bottom panel).

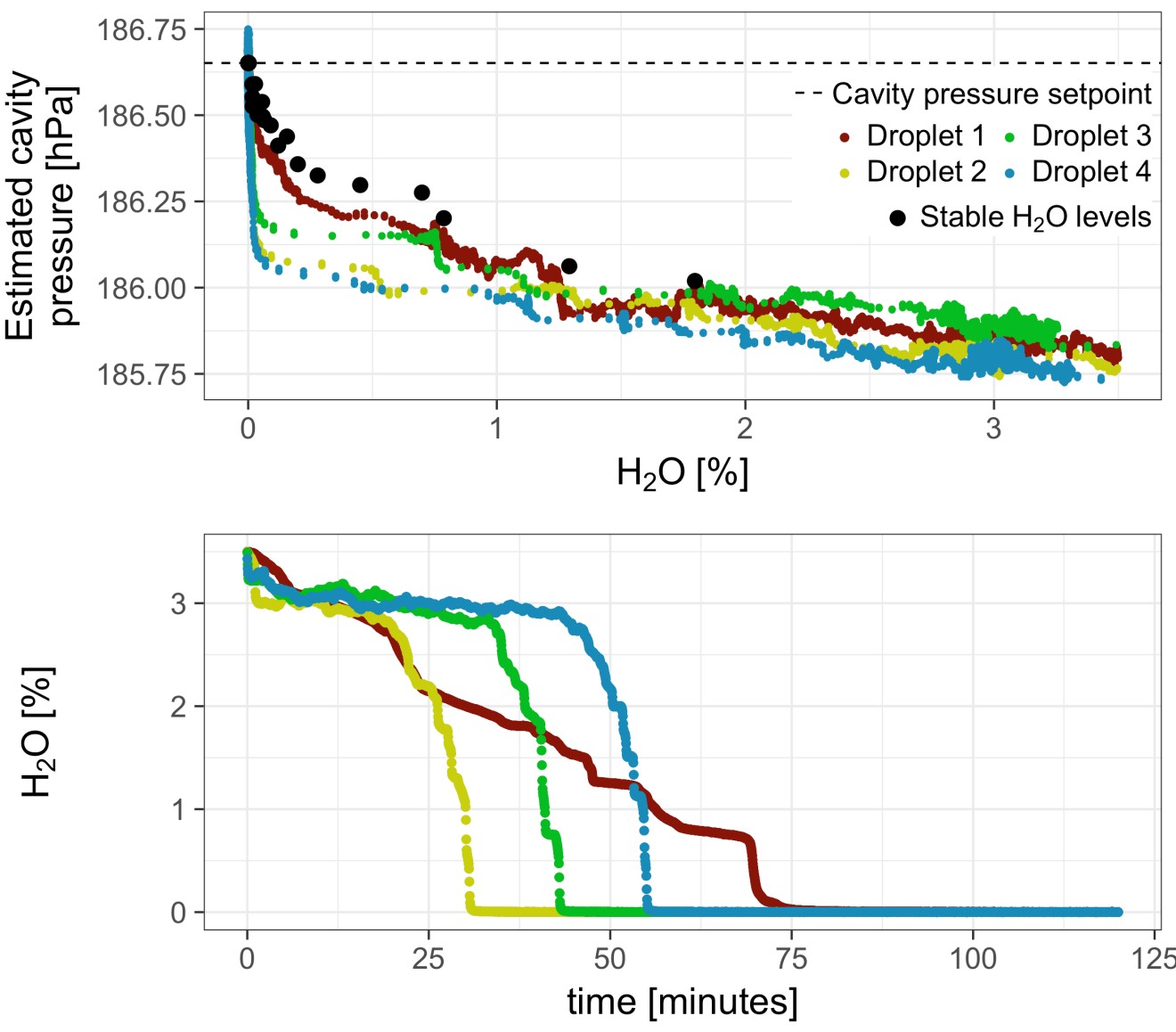

**Fig. 5: Top: Cavity pressure during droplet experiments with Picarro #1 estimated based on data from the external pressure sensor. For reference, the results from the experiment with stable water vapor levels from this analyzer are plotted as well (same as in Fig. 4). Bottom: Temporal progression of water vapor mole fraction during the droplet experiments after the drop below 3.5 % $H_2O$.**

### 3.2.3 Results from spectroscopic cavity pressure measurements

In the experiment with the oxygen analyzer (Sect. 2.6), $O_2$ line width measurements obtained for the same humidity levels throughout all cycles were stable (not shown). To reduce their noise, they were averaged over periods of 100 seconds. By contrast, the optical phase length of the cavity drifted over the course of the experiment (explained in Sect. S2). Therefore,

the averaged data based on the phase length were binned for further analysis, separately for the cycles between 0 and 0.2 % $H_2O$ and those between 0 and 0.8 % $H_2O$.

At water vapor mole fractions above 0.2 % $H_2O$, cavity pressure estimates based on optical phase length and $O_2$ line width both showed linear dependencies on water vapor, potentially with a small nonlinear component in the $O_2$ line width data
5   (Fig. 6). The linear dependencies can be ignored here, as they are compounded by effects other than cavity pressure changes (Sect. 2.3.2). Below about 0.2 % $H_2O$, both estimates exhibited the pressure bend that was also observed with the external pressure sensor. Fitting the empirical cavity pressure model Eq. (4) yielded coefficients for pressure bend position and magnitude very similar to those derived from data of the external pressure sensor (Table 4) and coefficients of determination larger than 0.98, which indicates good fits.

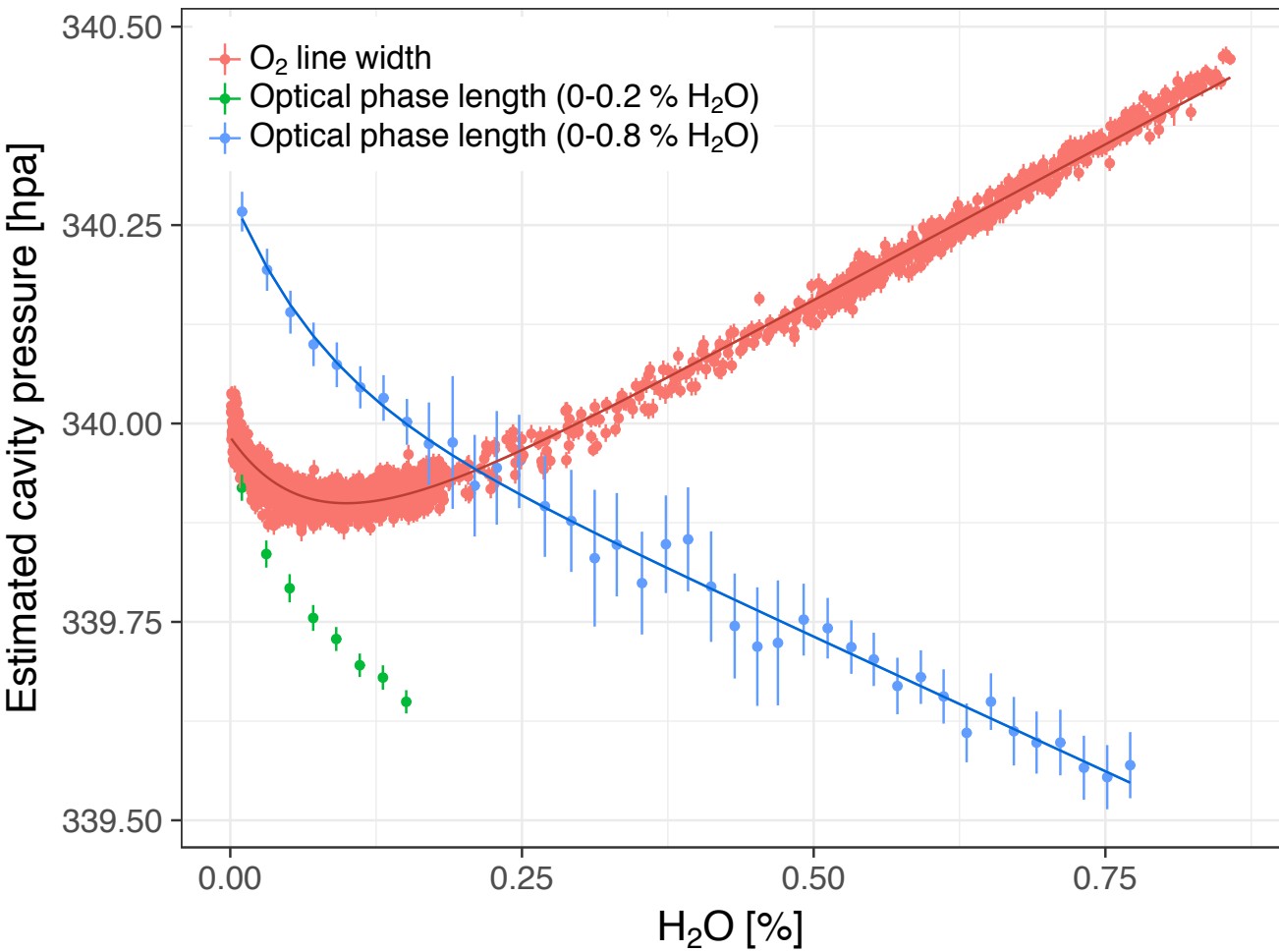

**Fig. 6: Cavity pressure estimated based on spectroscopic pressure measurements with Picarro #6 and fits of Eq. (4). Error bars of $O_2$ line widths and optical phase lengths are the standard errors of averaging and binning, respectively. Since the cycles up to 0.2 % $H_2O$ did not extend into the linear domain, the model was not fitted to the optical phase length data of these cycles. The slopes of the linear parts of the curves are compounded by other effects than cavity pressure variations (see Sect. 2.3.2).**

**Table 4: Coefficients for the empirical cavity pressure model Eq. (4) based on spectroscopic methods (estimates and standard errors). The last line shows averages and, as uncertainty, half the spreads of the individual estimates. The average of the slopes is not given because the slopes are caused by different physical processes.**

| Method | $s$ [hPa (% $H_2O$)$^{-1}$] | $h_p$ [% $H_2O$] | $d_p$ [hPa] |
|---|---|---|---|
| $O_2$ line width | $0.443 \pm 0.002$ | $0.076 \pm 0.002$ | $0.221 \pm 0.002$ |
| Optical phase length | $-0.38 \pm 0.02$ | $0.078 \pm 0.019$ | $0.222 \pm 0.024$ |
| Average | - | $0.0767 \pm 0.0008$ | $0.2216 \pm 0.0006$ |

### 3.3 Modification of standard water correction model to account for cavity pressure sensitivity to water vapor

Based on the results from sensitivity experiments and independent cavity pressure measurements, the standard water correction model Eq. (3) was modified to account for cavity pressure sensitivity to water vapor. First, the impact of measured deviations of cavity pressure from its nominal value ($\Delta p = p - p_0$) was subtracted from the wet air mole fractions. Then, the

10 standard water correction model was applied to the modified wet air mole fractions:

$$c_{dry}^{pressure-correction} = \frac{c_{wet}(h) - \frac{\partial c}{\partial p} \cdot \Delta p}{f_c^{para}(h)} \tag{5}$$

Here, $\frac{\partial c}{\partial p}$ is the sensitivity of the trace gas to cavity pressure changes. Henceforth, we call this the "pressure-correction" model.

The pressure-correction model requires independent measurements of cavity pressure. To eliminate the need for such measurements, the model was reformulated based on the empirical pressure correction model by substituting $\Delta p$ in Eq. (5)

15 with ($p_{est} - p_0$) from Eq. (4) and rearranging the terms, which yields

$$c_{dry}^{expanded} = \frac{c_{wet}(h)}{f_c^{exp}(h)} \tag{6}$$

with an "expanded" water correction function $f_c^{exp}(h)$:

$$f_c^{exp}(h) = \underbrace{1 + a_c \cdot h + b_c \cdot h^2}_{f_c^{para}(h)} + d_c \cdot \left( e^{-\frac{h}{h_p}} - 1 \right) \tag{7}$$

Here, $h_p$ is the pressure bend position from Eq. (4), and $d_c = d_p \cdot \frac{\partial c}{\partial p}$. Possible sensitivity of $\frac{\partial c}{\partial p}$ to water vapor, which was not detected in sensitivity experiments (Sect. 3.1), was neglected here. Coefficients for this model can be estimated from trace gas data, i.e. independent cavity pressure measurements are not needed.

## 3.4    Water corrections based on experiments with stable water vapor levels

### 3.4.1    Experiment with external pressure measurement

In this section, we show biases of the standard water correction model and link them to the cavity pressure sensitivity to water vapor. For this purpose, we collected data for both cavity pressure and the target gases $CO_2$ and $CH_4$ in one stable water vapor level experiment (with Picarro #3). We compare dry air mole fractions based on the standard, pressure-correction and expanded water correction models (Eq. (3), (5) and (6), respectively). In Fig. 7, we present dry air mole fractions alongside the WMO internal reproducibility goals. This context was chosen because, as stated in Sect. 1, keeping the bias of an individual measurement system between calibration scale and measurement within these goals ensures achieving the inter-laboratory compatibility goals.

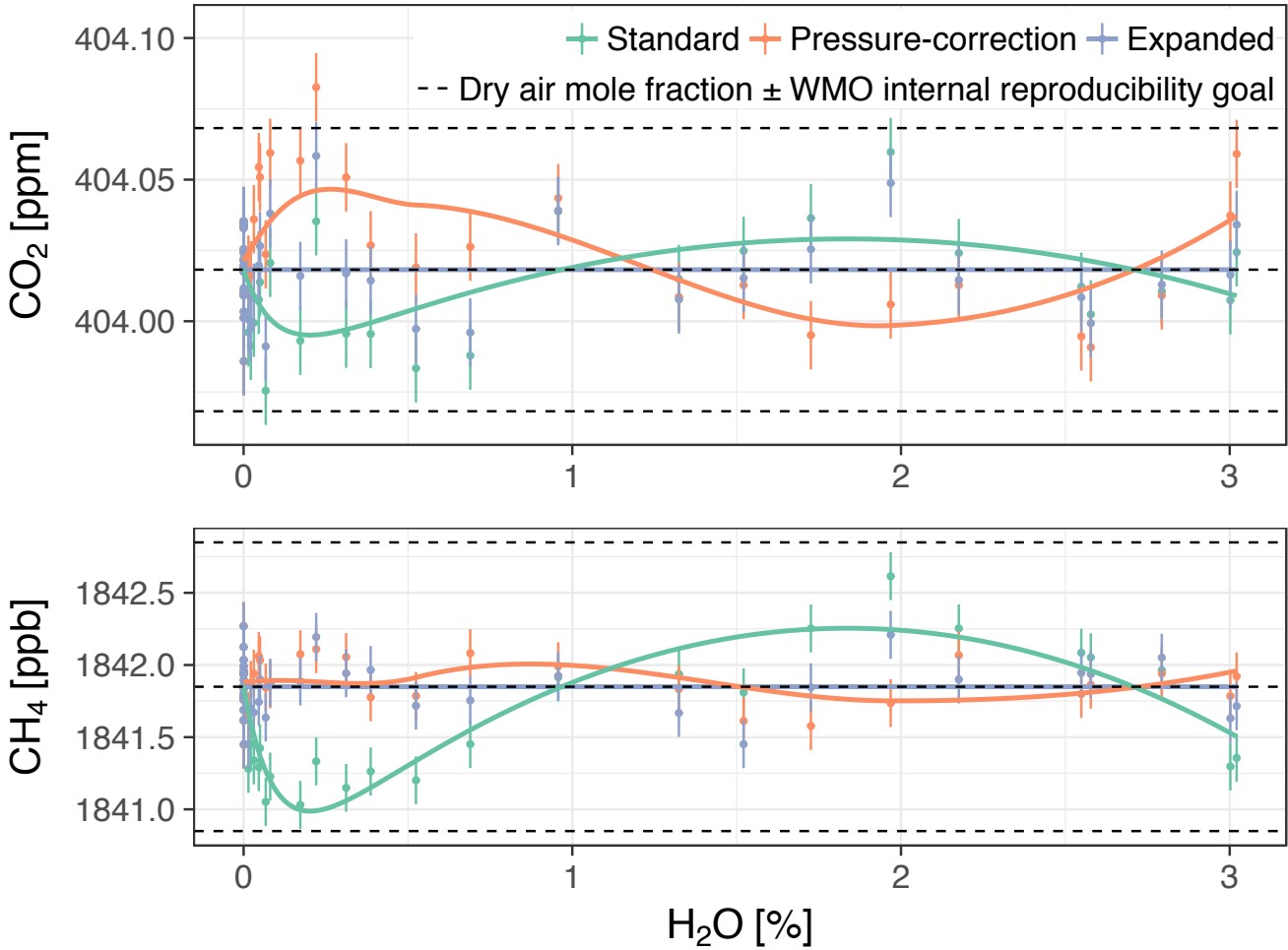

Fig. 7: Dry air mole fractions from the experiment with Picarro #3 based on standard water correction model, pressure correction model (i.e. using independently measured cavity pressure) and expanded water correction model (i.e. using the empirical dependence of cavity pressure on water vapor). Error bars: one standard deviation of the trace gas mole fractions measured in dry

air. The solid lines are the biases of the models assuming the expanded model was unbiased (smoothed for the pressure-correction model), offset by the mole fractions measured in dry air. The upper and lower dashed lines correspond to the WMO internal reproducibility goals (see Sect. 3.4.1), in the case of $CO_2$ in the northern hemisphere (WMO, 2016).

Dry air mole fractions of $CH_4$ calculated using the standard water correction model had a water-dependent structure (Fig. 7, bottom panel), with sustained negative biases at water vapor levels below 1 % $H_2O$ as the most prominent feature. This structure was eliminated by the pressure-correction and the expanded model, so that the dry air mole fractions based on these models varied less (Table 5). The largest difference between standard and expanded water correction model occurred at 0.2 % $H_2O$ (Table 6). Differences between pressure-correction and standard model were small (Fig. 7, bottom panel).

For $CO_2$, dry air mole fractions based on the standard model had a similar structure as the $CH_4$ mole fractions, but the differences to the expanded water correction model, which performed best, were much smaller than for $CH_4$ in terms of the overall variability (Table 5) and compared to the WMO internal reproducibility goals in the northern hemisphere (Fig. 7, top panel and Table 6). The pressure-correction model showed a comparatively poor performance, dominated by a small bias similar to the one present in the results of the standard model but with opposite sign (Fig. 7, top panel).

**Table 5: Standard deviations of dry air mole fractions based on different water correction models from the experiment with Picarro #3.**

| Model | St. dev. $CO_2$ | St. dev. $CH_4$ |
|---|---|---|
| Standard | 0.017 ppm | 0.35 ppb |
| Pressure-correction | 0.019 ppm | 0.16 ppb |
| Expanded | 0.014 ppm | 0.18 ppb |

**Table 6: Maximum differences between dry air mole fractions based on standard and expanded water correction model from the**
**experiment with Picarro #3. The largest differences are also given as percentages of the mole fractions measured in dry air.**

| | $CO_2$ | $CH_4$ | Position |
|---|---|---|---|
| Negative | 0.023 ppm / 0.006 % | 0.86 ppb / 0.047 % | 0.2 % $H_2O$ |
| Positive | 0.011 ppm | 0.41 ppb | 1.7 % $H_2O$ |
| Range | 0.034 ppm | 1.27 ppb | |

### 3.4.2 Variability between experiments with the same analyzer

With Picarro #5, one gas washing bottle experiment was performed in each 2015 and 2017, without external cavity pressure monitoring. In the 2015 experiment, the number of data points was insufficient to fully constrain both $h_p$ and $d_c$ in the
expanded water correction model. Since the (uncertain) estimate of $h_p$ based on $CH_4$ was close to the mean of $h_p$ from the

three experiments with external cavity pressure monitoring ($h_p^{mean} = (0.079 \pm 0.014)$ % $H_2O$), $h_p$ was set to $h_p^{mean}$ for this experiment. We also considered using $h_p$ from the 2017 experiment instead, but this induced biases in water-corrected $CH_4$ mole fractions. For the 2017 experiment, the estimate of $h_p$ based on $CH_4$ data was used also for $CO_2$, because its estimate based on $CO_2$ data was highly uncertain.

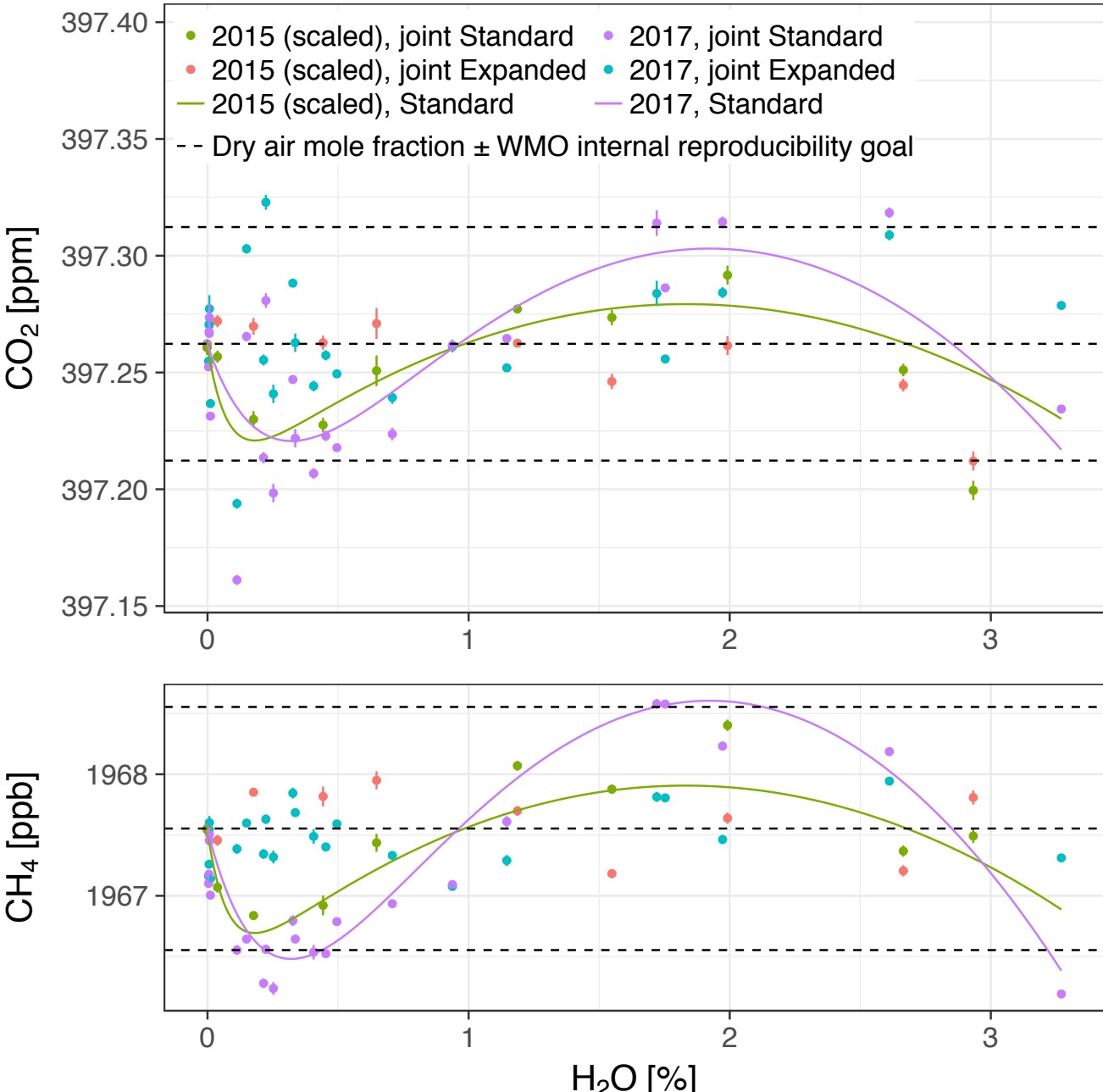

**Fig. 8: Water-corrected dry air mole fractions from the two experiments with Picarro #5. The data from the 2015 experiment have been scaled up to match the mole fractions measured in dry air in the 2017 experiment. The points are based on model fits to data from both experiments jointly (error bars: lower bounds of uncertainty; see Sect. S3), while the solid lines show differences between standard and expanded water correction model fitted to data from the 2015 and the 2017 experiments individually, offset by the mole fractions measured in dry air in the 2017 experiment. The dashed lines are the same as in Fig. 7.**

For both experiments, dry air mole fractions of $CO_2$ and $CH_4$ obtained using the standard water correction model had negative biases around the pressure bend position and at the highest sampled water vapor mole fractions (3 % $H_2O$), and a positive bias in between (lines in Fig. 8, Table 7). The biases were eliminated by the expanded model (Table 7). The magnitudes of the biases of water-corrected $CO_2$ mole fractions were consistent with those of $CH_4$. In the 2015 experiment, the largest bias occurred around the pressure bend position, while in the 2017 experiment, the largest positive biases, which occurred at 1.9 % $H_2O$, and the negative biases at the highest sampled water vapor mole fractions were on par with those at the pressure bend position (Table 7). Residuals were much larger than the estimated lower bounds of the uncertainty (error bars in Fig. 8), owing to the fact that not all uncertainties could be quantified (Sect. S3).

**Table 7: Comparison of water corrections of the two experiments with Picarro #5. The bias estimates of the standard model are based on the assumption that the results of the expanded model were unbiased.**

| | $CO_2$ | | $CH_4$ | |
|---|---|---|---|---|
| | 2015 experiment | 2017 experiment | 2015 experiment | 2017 experiment |
| *Coefficients (mean ± SE) (individual experiments)* | | | | |
| $h_p$ | See $CH_4$ | See $CH_4$ | $(0.079 \pm 0.014)$ % $H_2O$ (from Table 3) | $(0.26 \pm 0.06)$ % $H_2O$ |
| $d_c$ | $(1.6 \pm 0.3) \times 10^{-4}$ | $(3.0 \pm 0.6) \times 10^{-4}$ | $(6.6 \pm 1.1) \times 10^{-4}$ | $(1.7 \pm 0.1) \times 10^{-3}$ |
| *Coefficients (mean ± SE) (joint correction with data from both experiments)* | | | | |
| $h_p$ | See $CH_4$ | | $(0.16 \pm 0.04)$ % $H_2O$ | |
| $d_c$ | $(2.3 \pm 0.4) \times 10^{-4}$ | | $(1.19 \pm 0.08) \times 10^{-3}$ | |
| *Standard deviations (individual experiments and joint correction)* | | | | |
| Standard model | 0.02 ppm | 0.04 ppm | 0.39 ppb | 0.7 ppb |
| Expanded model | 0.01 ppm | 0.02 ppm | 0.17 ppb | 0.2 ppb |
| Expanded model (joint correction) | 0.016 ppm | 0.027 ppm | 0.24 ppb | 0.23 ppb |
| *Maximum biases of the standard model assuming the expanded model was unbiased (individual experiments)* | | | | |
| Negative, position (< 1 % $H_2O$) | 0.037 ppm/0.0104 %, 0.18 % $H_2O$ | 0.041 ppm/0.0105 %, 0.32 % $H_2O$ | 0.78 ppb/0.0437 %, 0.18 % $H_2O$ | 1.07 ppb/0.0545 %, 0.32 % $H_2O$ |
| Positive, position | 0.015 ppm, 1.8 % $H_2O$ | 0.043 ppm, 1.9 % $H_2O$ | 0.32 ppb, 1.8 % $H_2O$ | 1.10 ppb, 1.9 % $H_2O$ |
| *Maximum differences by swapping coefficients of expanded model between individual experiments* | | | | |
| < 1 % $H_2O$ | 0.02 ppm/0.049 % | | 0.6 ppb/0.030 % | |
| > 3 % $H_2O$ | 0.07 ppm/0.018 % | | 0.4 ppb 0.022 % | |

The water correction coefficients obtained from the two experiments had significant differences (Table 7). To assess the impact of these differences on water-corrected dry air mole fractions, two analyses were performed. First, the coefficients of either experiment were applied to the other one. This resulted in differences around the pressure bend positions, but they were smaller than the differences between standard and expanded water correction model. In addition, $CO_2$ differed at the largest water vapor mole fraction sampled (Fig. 8, top panel, Table 7). For a second assessment of differences between the two experiments, the 2015 data was scaled up to the mole fractions measured in dry air in the 2017 experiment and the expanded model was fitted to all data to obtain joint water corrections (points in Fig. 8). Standard deviations of the water-corrected dry air mole fractions based on the joint correction were between those based on the individual standard and expanded models (Table 7).

### 3.4.3    A case without bias of the standard water correction model

With Picarro #4, a gas washing bottle experiment without independent cavity pressure monitoring was performed. Dry air mole fractions obtained with the standard water correction model did not exhibit the systematic biases observed in Picarros #3 and #5 (Fig. 9) and had standard deviations of 0.016 ppm $CO_2$ and 0.21 ppb $CH_4$. This is better than the performance of the standard model in the experiments with the other analyzers, and for $CH_4$ close to the performances of the expanded model. Applying the expanded model to these data yielded insignificantly small pressure bend magnitudes $d_c$ and thus very similar dry air mole fractions without improvement of the variability (not shown). Residuals were much larger than the estimated lower bounds of the uncertainty (error bars in Fig. 9), owing to the fact that not all uncertainties could be quantified (Sect. S3).

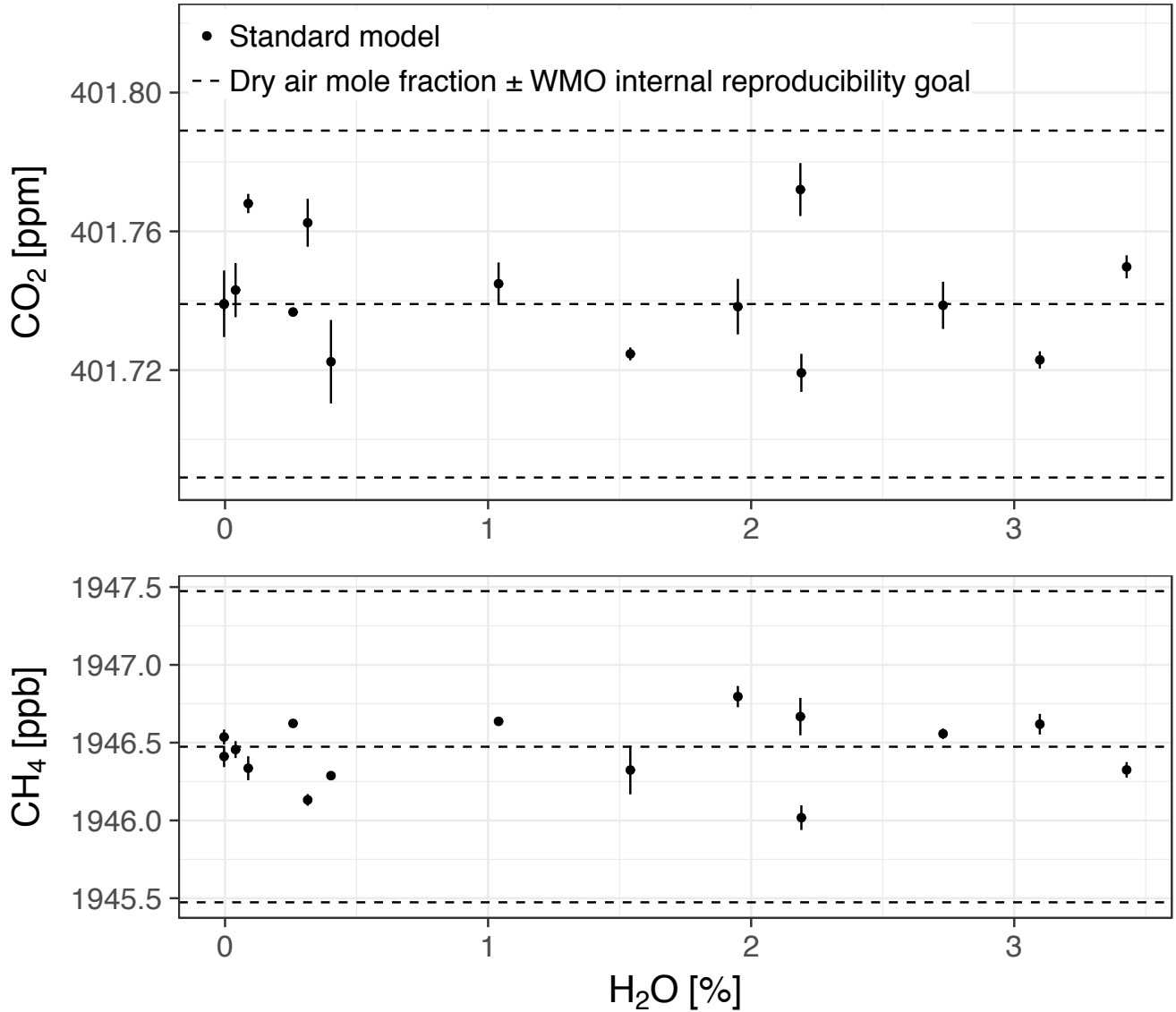

**Fig. 9: Dry air mole fractions of $CO_2$ and $CH_4$ for a gas washing bottle experiment with Picarro #4 based on the standard water correction model. Error bars: lower bound of uncertainty; see Sect. S3. The dashed lines are the same as in Fig. 7.**

5    3.5    **Water corrections based on droplet experiments**

The water correction models were fitted to the data from droplet experiments where the water vapor mole fraction was below 3.5 % $H_2O$ and where the difference between subsequent $H_2O$ measurements was smaller than 0.005 % $H_2O$. The former filter ensured compatibility with the gas washing bottle experiments, while the latter was an empirical filter to exclude the

fastest water vapor variations, which resulted in large variations of $CO_2$ and $CH_4$ readings, while leaving enough data for fitting.

Dry air mole fractions obtained with the standard water correction model had the typical bias structure that was also observed during gas washing bottle experiments (compare Fig. 10 with Fig. 7 and Fig. 8). Both the pressure-correction and the expanded model reduced or eliminated the biases induced by the standard model, with better performance of the pressure-correction model (Table 8). While the $CH_4$ bias at low water vapor mole fractions was eliminated by the pressure-correction model, the bias of $CO_2$ was only reduced.

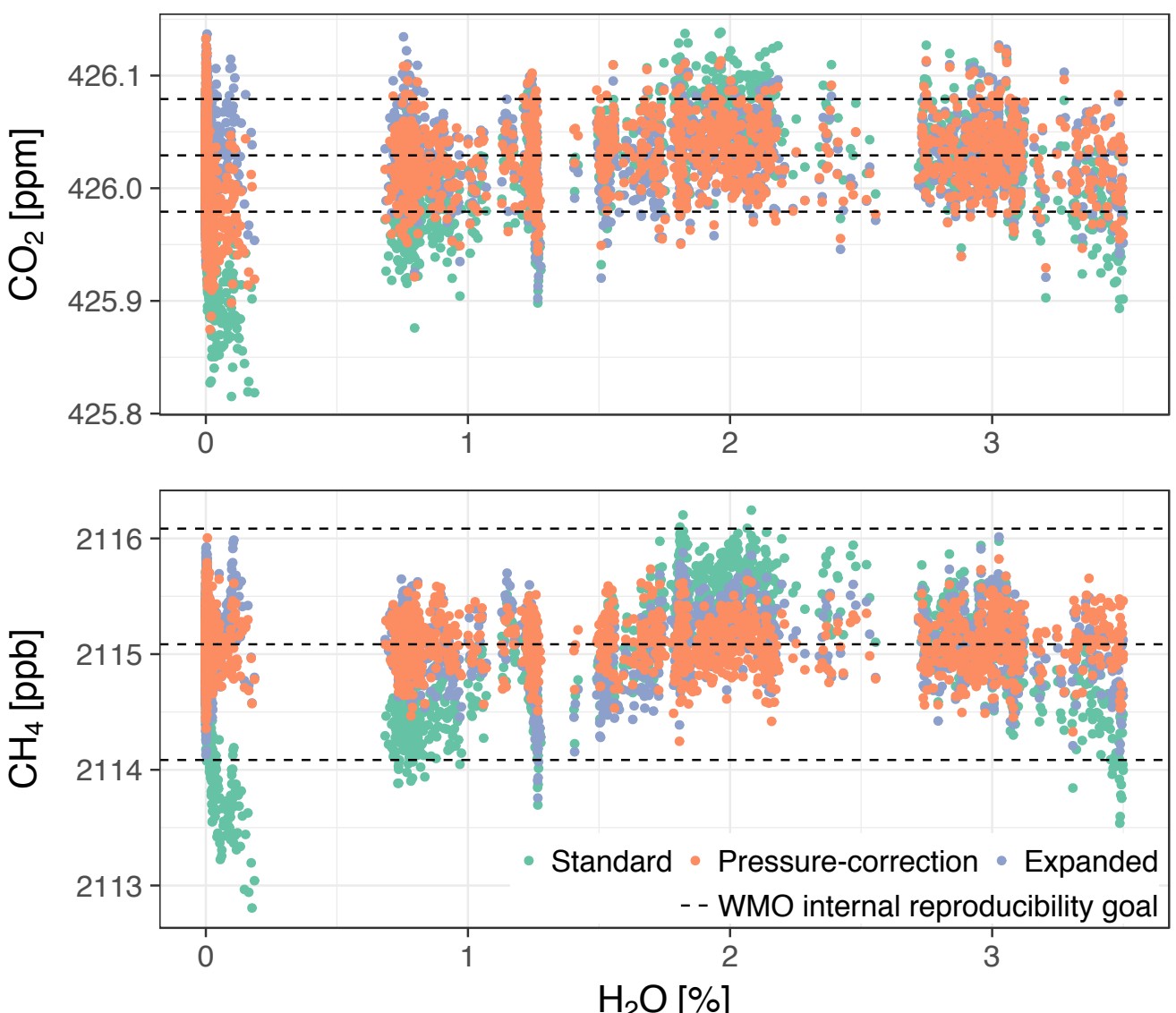

**Fig. 10: Dry air mole fractions from droplet experiment 1 with Picarro #1 based on the three water correction models. Droplet 1 is shown because it yielded the most data points after applying the filters described in the text. The dashed lines are the same as in Fig. 7.**

**Table 8: Average standard deviations of dry air mole fractions from all droplet experiments with Picarro #1 based on the three water correction models.**

| Model | St. dev. $CO_2$ | St. dev. $CH_4$ |
|---|---|---|
| Standard | 0.042 ppm | 0.42 ppb |
| Pressure-correction | 0.036 ppm | 0.26 ppb |
| Expanded | 0.039 ppm | 0.35 ppb |

During the fast decreases of water vapor mole fractions from about 0.5–1 % to 0 % $H_2O$ (Sect. 3.2.2), differences between wet air mole fractions between droplet experiments were large. The differences were quantified based on fitting the water correction functions of all models to wet air mole fractions from the individual droplets. The expanded function captured the large differences, which were up to 0.17 ppm $CO_2$ and 6.0 ppb $CH_4$ (Fig. 11). By contrast, differences between fits of the parabolic water correction function to wet air mole fractions (standard model), as well as to pressure-corrected wet air mole fractions (pressure-correction model), were much smaller, i.e. 0.04 ppm $CO_2$ and 0.8 ppb $CH_4$ (not shown).

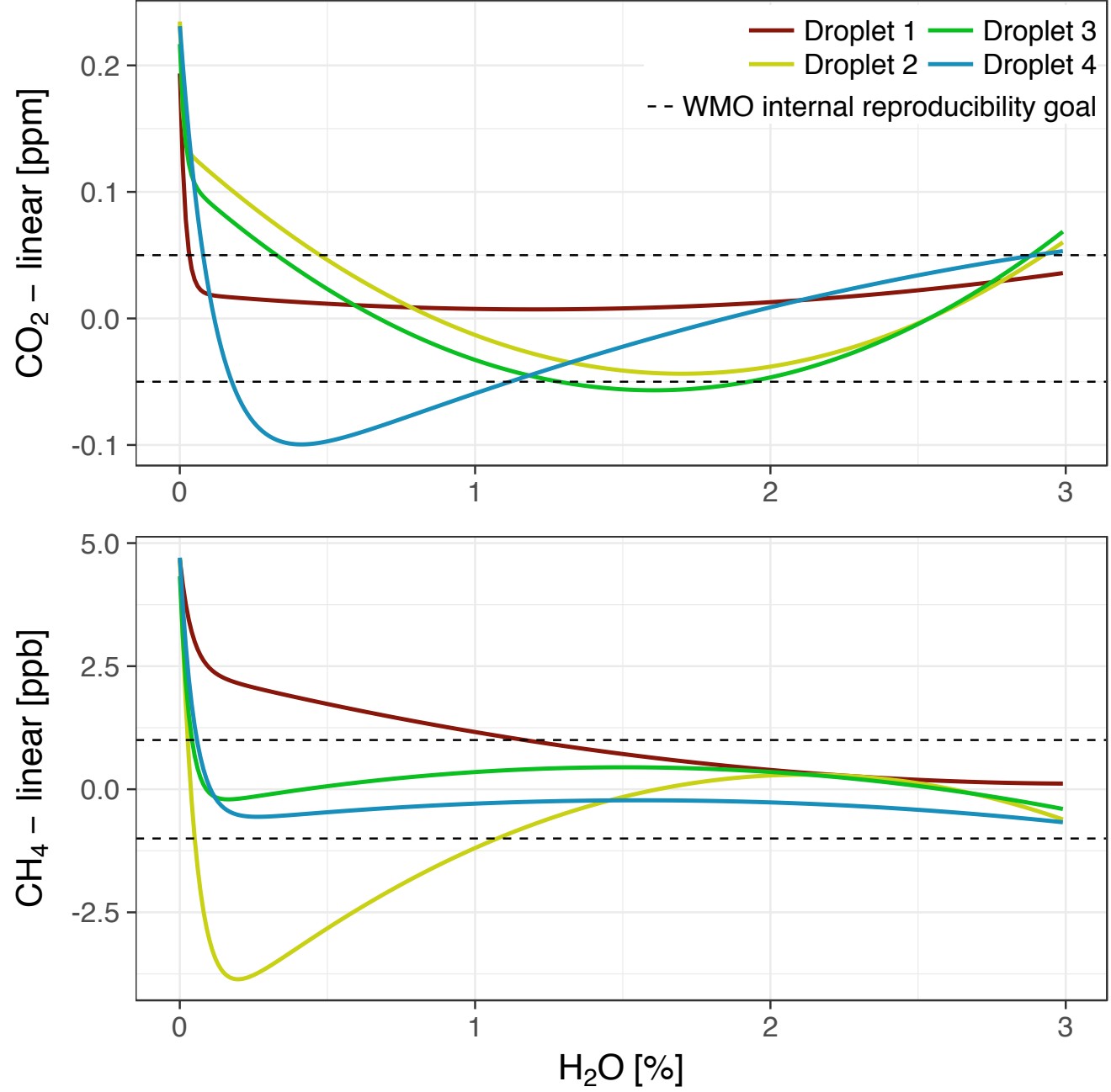

Fig. 11: Expanded water correction model fitted to data from four droplet experiments with Picarro #1. To emphasize the large differences, a common linear component has been subtracted. The dashed lines are the same as in Fig. 7.

# 4    Discussion

## 4.1    Findings from sensitivity experiments

Sensitivity experiments revealed sensitivities of $CO_2$ and $CH_4$ readings of Picarro GHG analyzers to cavity pressure. This demonstrates that trace gas readings are affected by systematic biases of cavity pressure.

Furthermore, these sensitivity experiments established the ability of our independent cavity pressure monitoring methods to detect cavity pressure changes. As a caveat, the sensitivity experiments did not characterize potential direct sensitivities, unrelated to cavity pressure changes, of the independent pressure monitoring methods to water vapor changes. For the approach using the external pressure sensor, the experiments were designed to prevent such sensitivity by installing the sensor behind a drying cartridge and in a dead end. Nonetheless, several parts of the setup may have caused a sensitivity of

the readings of the external sensor to water vapor changes (details are given in Sect. S1). In the approach using spectroscopic pressure measurements, experiments with varying water vapor indeed revealed linear dependencies on water vapor. Since their sign differed, they must at least partly have been caused by other effects than cavity pressure changes (Fig. 6). However, linear dependencies of the independent pressure estimates on water vapor do not affect our conclusions, since they are covered by the water correction models. The key result of our experiments, the pressure bend, was broadly consistent

between data from the external pressure sensor, both spectroscopic cavity pressure estimates, and $CH_4$ data. Given that all of these quantities were estimated based on different, unrelated methods, it is unlikely that our independent cavity pressure monitoring methods had systematic, water-dependent biases that affected our conclusions.

## 4.2    Cavity pressure of Picarro analyzers is sensitive to water vapor

Results from all independent cavity pressure measurements demonstrate that cavity pressure of Picarro analyzers is sensitive

to the water vapor content of the sample air. We described the sensitivity empirically based on the results of experiments with stable water vapor levels and external cavity pressure monitoring with Eq. (4).

Results from either humidification method indicate that cavity pressure takes time to adjust to new water vapor levels. To investigate whether cavity pressure equilibration affected the conclusions drawn from water correction experiments with stable water vapor levels, we inspected long (5–12 hours) measurements of dry air after switching from humid air for

evidence of cavity pressure equilibration longer than our typical probing time of humid air (40 minutes) and found only small variations (Sect. S1.1). We did not check for long equilibration after switching from dry to humid air. However, both in gas washing bottle and in droplet experiments, there was no indication that cavity pressure equilibrated more slowly with increasing than with decreasing water vapor mole fraction. Therefore, it is unlikely that cavity pressure equilibration affected the conclusions drawn from the experiments with stable water vapor levels.

Results from spectroscopic cavity pressure measurements agreed with the results of the external pressure sensor. Both the estimate based on $O_2$ line width and the one based on optical phase length exhibited the pressure bend with the same sign, and at a position and magnitude close to the average of the estimates based on the external pressure sensor. We note that we

expected the magnitude of the pressure bend to scale with cavity pressure, i.e. that it would be larger than estimates based on GHG analyzers by a factor of $\approx 1.8$, the ratio of cavity pressures of these instruments. Given the variability of $d_p$ between the three experiments with GHG analyzers, it is not certain whether this was the case.

We speculate that the observed sensitivity of internal pressure readings to humidity levels in sampled air is due to adsorption of $H_2O$ molecules on the pressure sensor inside the cavity. The pressure measurement is based on a piezoresistive strain gauge exposed to the pressure media (air in the cavity). The strain gauge is mounted on a thin diaphragm, which is deflected by pressure. The resulting strain causes a change in electrical resistance and creates an output voltage varying with pressure. Water molecules adsorbed on the strain gauge, diaphragm, or adjacent parts of the sensor may change its response to pressure mechanically, and/or may affect the electrical properties of the circuit. However, elucidating the underlying physical effect of the cavity pressure changes is beyond the scope of this paper and was not investigated further.

Since $CO_2$ and $CH_4$ readings react to changes in cavity pressure, the sensitivity of cavity pressure to water vapor affects $CO_2$ and $CH_4$ readings in humid air. Therefore, the results on cavity pressure imply that an adequate correction method is required to avoid systematic biases in water-corrected dry air mole fractions of $CO_2$ and $CH_4$ due to the cavity pressure dependence on water vapor.

### 4.3 Cavity pressure sensitivity to water vapor causes biases in $CO_2$ and $CH_4$ readings

Applying the standard water correction model resulted in biases in water-corrected $CO_2$ and $CH_4$ mole fractions in experiments with stable water vapor levels and droplet experiments. The shortcoming of the standard water correction model is that it is unable to model the pressure bend. The pressure-correction model, which directly links independently estimated cavity pressure to trace gas readings, eliminated the biases in $CH_4$ in all experiments. Although results for $CO_2$ were mixed (see Sect. 4.7), the performance of the pressure-correction model demonstrates a link between cavity pressure sensitivity to water vapor and trace gas readings of Picarro GHG analyzers in humid air. Biases of the standard model depend on the dry air mole fraction, and in our experiments amounted to up to 50 % of the WMO inter-laboratory compatibility goal for $CH_4$ and 80 % of the goal for $CO_2$ in the southern hemisphere (Picarro #5, 2017 experiment).

### 4.4 Correcting for cavity pressure sensitivity to water vapor without independent cavity pressure measurements

We developed the expanded water correction model to allow correction for the sensitivity of cavity pressure to water vapor without independent cavity pressure measurements. The model combined the parabolic water correction model from the literature with our empirical description of the dependency of cavity pressure on water vapor, which was composed of a linear term and an exponential term describing the pressure bend. We note that $O_2$ line width data suggest a small curvature of the cavity pressure dependency beyond the pressure bend (Fig. 6), as do data from the external pressure sensor during droplet experiments at water vapor mole fractions larger than those covered by our experiments with stable water vapor levels (Fig. 5, top panel). However, small curvatures can be captured by the parabolic part of all models, implying the expanded model is suitable despite potential shortcomings of the empirical cavity pressure model it was based on.

### 4.4.1 Experiments with stable water vapor levels

In the water correction experiment with stable water vapor levels and external cavity pressure monitoring, the $CH_4$ results of the expanded model closely matched those of the pressure-correction model (Sect. 3.4). It also fitted the observed $CO_2$ mole fractions from this experiment well, but their inconsistency with data from the external pressure sensor puts these $CO_2$ data into question (Sect. 4.7). More water correction experiments with stable water vapor levels were performed without independent cavity pressure measurement. In these experiments, consistency with cavity pressure could not be checked directly, but comparing the pressure bend magnitudes $d_{CO_2}$ and $d_{CH_4}$, as well as estimates of $h_p$ based on either trace gas provides useful information on potential inconsistencies. For instance, in the experiments with Picarro #5, $d_{CO_2}$ and $d_{CH_4}$ were broadly consistent (not shown), while in the experiment with Picarro #3, $d_{CO_2}$ was smaller than expected. In conclusion, $CO_2$ and $CH_4$ readings can be corrected for the dependency of cavity pressure on water vapor based on experiments with stable water vapor levels using the expanded water correction model, which does not require independent cavity pressure monitoring. Water correction experiments need to sample water vapor mole fractions between 0 and 0.5 % $H_2O$ sufficiently densely to constrain the pressure bend.

### 4.4.2 Droplet experiments

During droplet experiments, cavity pressure depended on the temporal course of water vapor variation. In particular, water vapor diminished quickly around the pressure bend position, but with a different temporal course in each experiment. Cavity pressure estimated based on the external pressure sensor was lower than during the experiment with stable water vapor levels and at the same time inconsistent around the pressure bend position, with the slowest-evaporating droplet closest to the data from experiment with stable water vapor levels. This suggests that the fast water vapor variations did not allow the measurements of the internal cavity pressure sensor to equilibrate, which caused biased $CO_2$ and $CH_4$ readings. While the biases were mitigated by the pressure-correction model, applying the expanded model yielded exaggerated and inconsistent pressure bends. Therefore, the results of our droplet experiments proved unsuitable for correcting cavity pressure-related biases of $CO_2$ and $CH_4$ readings without independent cavity pressure monitoring. However, droplet 1 evaporated more slowly than the other droplets and the experiment yielded cavity pressure data closer to those from the experiment with stable water vapor levels. This experiment was performed on another day, and the setup was reassembled in between. Thus, the course of evaporation may have been affected by the length and shape of the tubing between droplet injection point and Picarro analyzer. Based on the results from this droplet, we speculate that droplet experiments with even slower evaporation may yield results from which coefficients for the expanded water correction model can be derived.

### 4.5 Temporal stability of expanded water correction model

With Picarro #5, two experiments with stable water vapor levels were performed two years apart. Coefficients of the expanded model differed significantly between these experiments. It is unclear whether the differences were due to limited

reproducibility, short-term variations or long-term drifts, and more experiments are required to understand the variability. Variability may also be caused by other mechanisms than the sensitivity of cavity pressure to water vapor, which may explain the differences at water vapor mole fractions well above the pressure bend position. The differences around the pressure bend position between the two experiments were smaller than biases of the standard model. Therefore, dry air mole

fractions in this domain based on either set of coefficients were likely more accurate than those based on the standard model despite the variation between the two experiments.

## 4.6   Differences of expanded water correction model between analyzers

In total, we performed water correction experiments with stable $H_2O$ levels for $CO_2$ and $CH_4$ with three Picarro GHG analyzers. While the position ($h_p$) and magnitude ($d_c$) of the pressure bend in $CO_2$ and $CH_4$ readings were broadly consistent

between Picarros #3 and #5 (with the exception that the effect on $CO_2$ of Picarro #3 appeared reduced; see Sect. 4.7), $CO_2$ and $CH_4$ readings from Picarro #4 exhibited no detectable pressure bend. The magnitude of the pressure bend of this analyzer may be smaller than that of the others, masked by random fluctuations, or not be present at all. Alternatively, the pressure bend position may have been at a higher water vapor level, so that the standard model could capture the bend. The differences between this analyzer and the others are not explained by estimated uncertainties (Sect. S3). Thus, they remain

an open question for future research. The differences imply that custom coefficients for the expanded model should be obtained for each Picarro analyzer.

## 4.7   Challenges for $CO_2$

In all water correction experiments with independent cavity pressure monitoring, $CO_2$ data were not fully consistent with independent cavity pressure data. In the water correction experiment with stable water vapor levels and external pressure

monitoring (Picarro #3), biases of dry air $CO_2$ mole fractions obtained using the standard water correction model were much smaller than expected from cavity pressure variations, i.e. the pressure-correction model overcompensated the bias of the standard model. By contrast, biases of dry air mole fractions of $CO_2$ obtained using the standard model based on data from droplet experiments were reduced by the pressure-correction model, but not fully eliminated. Since $CH_4$ data were consistent with data from the external pressure sensor (Sect. 4.3), the most likely cause for the mixed $CO_2$ results is variations of the

$CO_2$ mole fractions delivered to the analyzer. Since in all our water correction experiments the air stream was in contact with liquid water, the underlying reason may have been dissolution in and outgassing from these reservoirs. This would likely have affected $CO_2$ more than $CH_4$, since its solubility in water is much higher. During gas washing bottle experiments, we took this effect into account by carefully observing the equilibration of trace gas mole fraction readings. However, it is conceivable that our efforts were not sufficient. If this explanation were true, the systematic difference between dry air and

wet air $CO_2$ mole fractions in the experiment with Picarro #3 would have precisely compensated for the pressure bend, which seems unlikely. Therefore, we regard this interpretation with caution and acknowledge the possibility that another mechanism caused the inconsistencies of $CO_2$ readings with the data from the external pressure sensor (a more detailed

discussion can be found in Sect. S3). Overall, our results highlight the need for high quality data to correct $CO_2$ readings for the effects of water vapor.

## 5    Conclusions

We reported previously rarely detected and unexplained biases of $CO_2$ and $CH_4$ measurements obtained with Picarro GHG analyzers in humid air. They were largest at low water vapor mole fractions below 0.5 % $H_2O$, where they amounted to up to 50 % (~1 ppb) of the WMO inter-laboratory compatibility goal for $CH_4$, and 80 % (~0.04 ppm) for $CO_2$ in the southern hemisphere at ambient mole fractions.

The biases may not only affect measurements without drying systems, but also measurement systems that use Nafion membranes to dry air samples due to residual water vapor. Stavert et al. (2018) reported that in their setup, the Nafion membrane humidified calibration air to less than 0.015 % $H_2O$, while the humidity of the sample air was on average 0.2 % $H_2O$. This humidity difference could result in the maximum biases we observed. On the other hand, other studies reported smaller differences between the water levels of sample and calibration air after passing through Nafion (Verhulst et al., 2017; Welp et al., 2013). Eliminating differences between residual water vapor levels of sample and calibration air would remove the biases reported on here, as would drying sample air to very low water levels, e.g. using a cryotrap.

The biases are due to a sensitivity of the pressure in the measurement cavity to water vapor, which we observed both with an additional external pressure sensor, and based on spectroscopic methods. We speculate that the underlying physical mechanism of the cavity pressure variability is adsorption of water molecules on the piezoresistive pressure sensor in the cavity that is used to keep cavity pressure stable.

The biases can be corrected without independent cavity pressure measurements based on experiments with stable water vapor levels by an empirical expansion of the standard water correction model from the literature, which we derived from the cavity pressure dependency on water vapor.

Correction of the biases of $CO_2$ readings was challenging, presumably because of dissolution in and outgassing from the water reservoir used to humidify the air stream.

The commonly used droplet method did not yield results suitable for correcting biases of $CO_2$ and $CH_4$ readings related to cavity pressure without independent cavity pressure monitoring. In these experiments, water vapor varied faster than it takes cavity pressure to adjust to a new water vapor level. We speculate that water droplets may nonetheless be suitable for deriving coefficients for the expanded water correction model under the condition that evaporation is sufficiently slow. Since our results do not determine the necessary equilibration time, we recommend using humidification methods that allow maintaining stable water vapor levels. Since the humidification via gas washing bottle is complicated to implement in the field and may have affected our $CO_2$ results, alternative humidification methods may be more suitable. For example, Winderlich et al. (2010) achieved stable water vapor levels with much smaller amounts of liquid water in the air stream using a so-called "water trap", which is akin to a droplet experiment with more controlled evaporation.

Future research is necessary to understand differences of cavity pressure-related biases of $CO_2$ and $CH_4$ between analyzers and over time. Therefore, coefficients for the expanded model should be obtained for each analyzer individually, and be monitored over time.

The biases addressed here are on the order of magnitude of the WMO inter-laboratory compatibility goals. They did not exceed them, but several other error sources that affect GHG measurements, like tracing the calibration of the gas analyzer to a common primary scale (e.g. Andrews et al., 2014), are on the same order of magnitude. Therefore, to reach the WMO inter-laboratory compatibility goals, biases from each individual error source need to be "as small as possible" (Yver Kwok et al., 2015). Thus, accounting for cavity pressure-related biases of $CO_2$ and $CH_4$ readings contributes to keeping the compatibility of measurements performed with the widely used Picarro GHG analyzers in humid air and potentially in Nafion-dried air within the WMO inter-laboratory compatibility goals.

**Acknowledgements**

This work was supported by the Max-Planck Society, the European Commission (PAGE21 project, FP7-ENV-2011, grant agreement No. 282700; PerCCOM project, FP7-PEOPLE-2012-CIG, grant agreement No. PCIG12-GA-201-333796; INTAROS project, EU-H2020-BG-09-2016, grant agreement No. 727890), the German Ministry of Education and Research (CarboPerm-Project, BMBF grant No. 03G0836G), the AXA Research Fund (PDOC_2012_W2 campaign, ARF fellowship M. Göckede), and the European Science Foundation (TTorch Research Networking Programme, Short Visit Grant F. Reum). We thank Stephan Baum, Dietrich Feist and Steffen Knabe (MPI-BGC) for help with the experiments. We thank David Hutcherson (Amphenol Thermometrics (UK) Ltd) for clarifications regarding the piezoresisitive pressure measurement technique. We thank Andrew Durso, Dietrich Feist and Martin Heimann (MPI-BGC) for feedback on the manuscript.

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
