# Peer review of "Correcting atmospheric CO2 and CH4 mole fractions obtained with Picarro analyzers for sensitivity of cavity pressure to water vapor"

_Atmospheric Measurement Techniques, 2018_

## Referee Comment (RC1) · Anonymous Referee #1 · 18 Sep 2018

In this paper, the authors are presenting results aiming at correction CO2 and CH4 mole fractions for sensitivity of the cavity pressure to water vapor that induces an additional bias on top of the sensitivity on both gases to water vapor itself. It allows to correct for this previously observed bias in the low amount of water vapor. They propose methods to estimate the correction for each individual instrument and analyze the different sources of uncertainty. After minor corrections, the paper should be published.

p2 I5 networks

Fig 1 or 2 Please draw one of the set-up for the flight model and one for the other so the differences between the set-up are clearer.

p12 Fig6: why are the the coefficients so different between O2 line and optical phase length?

p14 Fig7: Why using equation 5 or 6 yields different results?

Table 7 is not fully clear with the text, I'm not sure where the expanded model biases are shown.

p19 I6 where the water vapor mole fraction was selected...

| Interactive comment on Atmos. Meas | . Tech. Discuss. | ., doi:10.5194/amt-2018-242, 201 | 8. |
|------------------------------------|------------------|----------------------------------|----|
|------------------------------------|------------------|----------------------------------|----|

---

## Referee Comment (RC2) · Anonymous Referee #2 · 24 Oct 2018

The manuscript "Correcting atmospheric CO2 and CH4 mole fractions obtained with Picarro analyzers for sensitivity of cavity pressure to water vapor" of F. Reum et al. studies the sensitivity of the pressure sensor to changes in water vapor of Picarro CRDS instruments. In some cases, this might be relevant, since changing humidity between calibration and measurement can affect the accuracy of the measurements.

The manuscript is well written and concise. The arguments are sound, and the topic is relevant for the community doing highly accurate measurements of CO2 and CH4. I therefore recommend publication in AMT after addressing the following concerns.

General comments

The paper is certainly relevant for users aiming at highest accuracy of their measurements. However, I miss a little bit the context to other potential sources of uncertainty. The effect seems to be small and less than the WMO compatibility goal. The proposed alternative water vapor correction will too complicated to implement for most users, and therefore the authors are encouraged to give guidance on how the effect can be avoided. This could be a recommendation that drying to very low humidity might be necessary if highest accuracy is required. I also think that a setup using Nafion dryers can be used if the calibration gases also pass over the dryer. The authors point out that the effect on the pressure sensor readings is also relevant for measurements made using a Nafion dryer. This certainly holds true if the calibration gas is not passing over the dryer. However, if the calibration gas is also passing over the Nafion dryer it will be humidified, which results in very small humidity changes between sample and calibration gas, and the effect might be neglected.

Could the Picarro software correct this "internally"? If all pressure sensors have the same or a similar water vapor dependent bias, a correction should be possible.

Section 3.1 is difficult to understand. I suggest adding a few words of explanation to the numbers given in Table 2, and discuss their meaning and relevance. Why are the water vapor readings not sensitive to pressure changes?

There is a large difference between droplet experiments shown in section 3.2.2. Experiments 2-4 shows a much faster decrease in H2O compared to experiment 1. Were the conditions different for those experiments?

The WMO compatibility goal is interpreted by the authors as a allowed bias of $\pm 0.05$ ppm for CO2 and $\pm 1.0$ ppb for CH4. However, the compatibility goals of WMO are a "maximum allowed bias", and should therefore be $\pm 0.1$ ppm for CO2 and $\pm 2.0$ ppb for CH4. Please correct this in the text and figures.

Specific comments

Page 4, lines 24-25: You state that the pressure of the external sensor was adjusted to be within a few hPa the same as inside the cavity by a needle valve. Please be more specific. How close was it? The optimal position for an external pressure measurement would be either between the cavity and the inlet or outlet valve, which would allow for the measurement of the same pressure as in the cavity without the influence of the loop feedback. Would that be feasible, and if yes, why was it not realized?

Page 5, line 31: What was the reason for the drift of the external pressure sensor? Could this be identified?

Page 18, section 3.4.3: Why is Picarro 4 performing better than others? Is it newer? Is it a different model (according to Table 1 G2401-mc; I could not find any information on a G2401-mc model on the Picarro website, only for G2401-m).

Technical corrections

Page 15, Table 6: Should the range for CH4 be 0.41 −(-0.86) = 1.27 (instead of 1.30)?.

Page 28, line 29: Reference of Stavert et al. is incomplete (journal is missing).

Page 29, line 5: Link to report is wrong.

---

## Author Comment (AC1) · 14 Nov 2018

We thank the referee for the comments on the discussion paper, which address minor issues. In the following, we address each comment.

Comment:
p2 l5 networks

Response:
To avoid confusion, we replace the phrase "are used at many sites of the international GHG monitoring network" with "are used at many GHG monitoring sites".

Comment:
Fig 1 or 2 Please draw one of the set-up for the flight model and one for the other so the differences between the set-up are clearer.

Response:
We will modify Fig. 1 and its caption accordingly.

Comment:
p12 Fig6: why are the the coefficients so different between O2 line and optical phase length?

Response:
We assume that the comment refers to the different signs of the slopes of the cavity pressure estimates. This was addressed in the text, but to make it clearer, we will include a reference to the corresponding text section in the figure caption:
"The slopes of the cavity pressure estimates based on spectroscopic methods differ because they are compounded by other effects (see Sect. 2.3.2)."

Comment:
p14 Fig7: Why using equation 5 or 6 yields different results?

Response:
The difference is that Eq. (5) makes use of independently measured cavity pressure (in these cases with the external pressure sensor), while Eq. (6) uses the empirical dependence of cavity pressure on water vapor. In other words, the difference is due to the noise of the independent cavity pressure measurement (assuming the empirical model is accurate). We will clarify this in the caption of the figure:
Before: "Dry air mole fractions from the experiment with Picarro #3 based on the three water correction models."
Corrected: "Dry air mole fractions from the experiment with Picarro #3 based on standard water correction model, pressure correction model (i.e. using independently measured cavity pressure) and expanded water correction model (i.e. using the empirical dependence of cavity pressure on water vapor).

Comment:
Table 7 is not fully clear with the text, I'm not sure where the expanded model biases are shown.

Response
The biases of the standard water correction model in Table 7 were based on the assumption that the expanded model was unbiased. This was stated in the caption of the table, but we will also add it to the section on the bias of the standard model.
Before: "Maximum biases of standard model (individual experiments)"
Revision: "Maximum biases of standard model assuming the expanded model was unbiased (individual experiments)"

Comment:
p19 l6 where the water vapor mole fraction was selected...

Response:
We will correct this typo as suggested.

---

## Author Comment (AC2) · 14 Nov 2018

We thank the referee for this thorough review, which uncovered a number of revisions needed to improve the manuscript.

General comments

Comment:
The paper is certainly relevant for users aiming at highest accuracy of their measurements. However, I miss a little bit the context to other potential sources of uncertainty. The effect seems to be small and less than the WMO compatibility goal.

Response:
The referee correctly points out that the cavity pressure dependence we address in this paper is only relevant for measurements aiming at the highest accuracy. Other measurement uncertainties are on the same order of magnitude as the biases we address, e.g. the accuracy of calibrations against the WMO scale (e.g. Andrews et al., 2014; Yver Kwok et al., 2015) or cylinder drift. Since the WMO inter-laboratory compatibility goals refer to compound errors from different sources of uncertainty, we regard any correction that is on the order of magnitude of the goals as a relevant contribution to the overall accuracy. Similarly, Yver Kwok et al. (2015) concluded that "to be able to reach the WMO comparison goals, we need biases as small as possible for every source of bias".
We will add this information to the conclusions:
Sect. 5:
Before: "Accounting for cavity pressure-related biases of CO2 and CH4 readings contributes to keeping the overall measurement uncertainty of the widely used Picarro GHG analyzers operated in humid air below the WMO inter-laboratory compatibility goals."
Revision: "The biases addressed here are on the order of magnitude of the WMO inter-laboratory compatibility goals, as are several other error sources that affect GHG measurements like tracing the calibration of the gas analyzer to a common primary scale (e.g. Andrews et al., 2014). Therefore, to reach the goals, biases from each individual source need to be as small as possible (Yver Kwok et al., 2015). Thus, accounting for cavity pressure-related biases of CO2 and CH4 readings contributes to keeping the overall measurement uncertainty of the widely used Picarro GHG analyzers operated in humid air below the WMO inter-laboratory compatibility goals".

Comment:
The proposed alternative water vapor correction will too complicated to implement for most users, and therefore the authors are encouraged to give guidance on how the effect can be avoided.

Response:
We comment on the possibility to avoid the effect below. However, we would also like to follow the referee's suggestion to provide an easier way to account for the pressure-related biases than our experiments. An alternative to our experiments could be to adjust the setup of water droplet experiments so that they can better maintain stable water vapor levels. This has been achieved with a so-called "water trap" by Winderlich et al.(2010), which we will reference in the conclusions:

Sect. 5:

Before: "The commonly used droplet method is not suitable for correcting biases of $CO_2$ and $CH_4$ readings related to cavity pressure without independent cavity pressure monitoring, because in these experiments, water vapor can vary faster than it takes cavity pressure to adjust to a new water vapor level."

Addition: "Since the humidification via gas washing bottle is complicated to implement in the field and may have affected our $CO_2$ results, alternative humidification methods may be more suitable. For example, Winderlich et al.(2010) achieved stable water vapor levels with much smaller amounts of liquid water in the air stream using a so-called "water trap", which is akin to a droplet experiment with more controlled evaporation."

See also the referee comment below on differences between droplet results for further modification of this section.

Comment:

This could be a recommendation that drying to very low humidity might be necessary if highest accuracy is required. I also think that a setup using Nafion dryers can be used if the calibration gases also pass over the dryer. The authors point out that the effect on the pressure sensor readings is also relevant for measurements made using a Nafion dryer. This certainly holds true if the calibration gas is not passing over the dryer. However, if the calibration gas is also passing over the Nafion dryer it will be humidified, which results in very small humidity changes between sample and calibration gas, and the effect might be neglected.

Response:

The pressure-related biases could be avoided by drying to very low humidities using a cryo trap. We will add this statement to the conclusions (see below).

We thank the referee for pointing out that, when employing Nafion, passing the calibration air through the membrane tube as well may resolve the humidity mismatch. However, even with this treatment, humidity differences can remain between sample and calibration air. Stavert et al. (2018) noted that their Nafion membrane humidified calibration air to less than 0.015 % $H_2O$, while the sample air averaged 0.2 % $H_2O$. This humidity difference would result in the maximum biases we observed. On the other hand, other studies reported smaller differences between the water levels of sample and calibration air after Nafion (Verhulst et al., 2017; Welp et al., 2013), which will considerably reduce the impact of the cavity-pressure related bias. Since we do not have much experience with the Nafion method, we will keep the conclusion that setups that employ Nafion may be affected, but will add that this will only be the case if sample and calibration air have different residual water vapor levels.

Abstract:

Before: "…and can therefore affect measurements obtained in humid air and in air dried with a Nafion membrane."

Revision: "…and can therefore affect measurements obtained in humid air. Setups that dry sample air using Nafion membranes may be affected as well if there are differences in residual water vapor levels of sample and calibration air."

Sect. 5:

Before: "As noted by Stavert et al. (2018), the biases may not only affect measurements without drying systems, but also measurement systems that use Nafion membranes to dry air samples, since the residual water vapor can be in the range of the where the largest biases occurred in our experiments (compare e.g. Verhulst et al., 2017)."

Revision: "Drying sample air to very low water levels, e.g. using a cryotrap, would eliminate the biases. However, the biases may affect measurement systems that use Nafion membranes to dry air samples due to residual water vapor. Stavert et al. (2018) reported that in their setup,

the Nafion membrane humidified calibration air to less than 0.015 % H2O, while the humidity of the sample air was on average 0.2 % H2O. This humidity difference could result in the maximum biases we observed. On the other hand, other studies reported smaller differences between the water levels of sample and calibration air after passing through Nafion (Welp et al., 2013; Verhulst et al., 2017). Eliminating differences between residual water vapor levels of sample and calibration air would remove the biases reported on here."

Comment:
Could the Picarro software correct this "internally"? If all pressure sensors have the same or a similar water vapor dependent bias, a correction should be possible.

Response:
Since we found as yet unexplained differences between instruments, we cannot give a "factory" correction of the pressure-induced biases.

Comment:
Section 3.1 is difficult to understand. I suggest adding a few words of explanation to the numbers given in Table 2, and discuss their meaning and relevance.

Response:
To improve clarity, we will exchange "slope" for "sensitivity". Furthermore, we will add an example to illustrate the magnitude:
Sect. 3.1:
Before: "… demonstrating that biases in cavity pressure directly affect mole fraction readings."
Addition: "For dry air mole fractions of 400 ppm CO2 and 2000 ppb CH4, a change of 1 hPa in cavity pressure makes a difference of 0.37 ppm CO2 and 6.4 ppb CH4 on average."

Comment:
Why are the water vapor readings not sensitive to pressure changes?

Response:
We think that the water vapor readings are also sensitive to cavity pressure, but that they were too variable in this experiment to detect the sensitivity based on our method. Since sensitivities of water vapor readings on cavity pressure that are undetectable at this level are irrelevant for the remainder of the paper, we did not investigate this further. To avoid confusion, we will remove the statement on the water vapor sensitivity from the manuscript.

Comment:
There is a large difference between droplet experiments shown in section 3.2.2. Experiments 2-4 shows a much faster decrease in H2O compared to experiment 1. Were the conditions different for those experiments?

Response:
Experiments 2–4 were made on another day than experiment 1, and the setup was reassembled in between. The course of water droplet evaporation may have been affected e.g. by length and shape of the tubing between Tee piece and Picarro analyzer. We will add this information to the text (see below).

In the submitted manuscript, we generalized our results that droplets do not provide stable enough water vapor levels for deriving coefficients for the expanded water correction model by concluding that the droplet method in general is not suitable. However, the differences brought up by the referee in this comment, in particular the fact that the slowest-evaporating droplet yielded results closer to those from the experiment with stable water vapor levels, suggest that droplets may yet be suitable, but under the condition that they yield water vapor levels that vary slowly enough. We will acknowledge this with the suggestion to use the water trap-method for air stream humidification, which may be regarded as a droplet experiment with controlled evaporation (see above). Therefore, we will remove the generalizations that water droplets are not suitable in our evaluations of droplet results:

Abstract:

Before: "The commonly used droplet method does not fulfill this requirement."

Revision: "In our experiments with the commonly used droplet method, this requirement was not fulfilled".

Sect. 4.4.2:

Before: "Therefore, the droplet method proved unsuitable for correcting cavity pressure-related biases of CO2 and CH4 readings without independent cavity pressure monitoring. "

Revision: "Therefore, the results of our droplet experiments proved unsuitable for correcting cavity pressure-related biases of CO2 and CH4 readings without independent cavity pressure monitoring. However, droplet 1 evaporated slower than the others and the experiment yielded cavity pressure data closest to those from the experiment with stable water vapor levels. This experiment was performed on another day, and the setup was reassembled in between. The course of evaporation may have been affected by the length and shape of the tubing between droplet injection point and Picarro analyzer. Based on the results from this droplet, we speculate that droplet experiments with even slower evaporation may yield results from which coefficients for the expanded water correction model can be derived."

Sect. 5:

Before: "The commonly used droplet method is not suitable for correcting biases of $CO_2$ and $CH_4$ readings related to cavity pressure without independent cavity pressure monitoring, because in these experiments, water vapor can vary faster than it takes cavity pressure to adjust to a new water vapor level."

Revision: "The commonly used droplet method did not yield results suitable for correcting biases of $CO_2$ and $CH_4$ readings related to cavity pressure without independent cavity pressure monitoring. In these experiments, water vapor varied faster than it takes cavity pressure to adjust to a new water vapor level. Cavity pressure during the experiment where the droplet evaporated slowest was closest to the data from the experiment with stable water vapor levels. Therefore, we speculate that water droplets may be suitable for deriving coefficients for the expanded water correction model, provided that evaporation is sufficiently slow. However, our results do not determine the necessary equilibration time. Therefore, we recommend using methods that allow maintaining stable water vapor levels."

See also the referee comment above on simpler ways to account for the pressure-related biases in the field for paragraph on the water trap method added to this section.

Comment:

The WMO compatibility goal is interpreted by the authors as a allowed bias of ±0.05 ppm for CO2 and ±1.0 ppb for CH4. However, the compatibility goals of WMO are a "maximum allowed bias", and should therefore be ±0.1 ppm for CO2 and ±2.0 ppb for CH4. Please correct this in the text and figures.

Response:

The thresholds in our figures refer to the WMO internal reproducibility goals, which we have not explicitly pointed out in the manuscript. The WMO inter-laboratory compatibility goals refer to maximum allowed biased between laboratories, not between laboratories and the calibration scale. Therefore, keeping biases of a laboratory below these goals does not ensure the same level of compatibility to other laboratories. If one laboratory has a $CO_2$ bias of +0.1 ppm, and another has a bias of -0.1 ppm, the relative bias between these laboratories is 0.2 ppm, exceeding the compatibility goal. However, keeping biases to the primary scale below half of the compatibility goals ensures achieving the compatibility goals between laboratories. Therefore, we used these thresholds in the figures in our manuscript. The WMO calls these thresholds the "internal reproducibility goals", which encompass "not only instrumental imprecision, but also uncertainties in transferring the calibration scale from the highest level of standards to working standards and other uncertainties, for example related to gas handling, at the field station or laboratory" (WMO, 2016).

We will add the information on the internal reproducibility goals to text and figure captions.

Sect. 1:

Before: "… the World Meteorological Organization (WMO) has set compatibility goals for atmospheric CO2 and CH4 measurements to ±0.1 ppm for CO2 (±0.05 ppm in the southern hemisphere) and ±2 ppb for CH4 (WMO, 2016)."

Revision: "… the World Meteorological Organization (WMO) has set compatibility goals for atmospheric CO2 and CH4 measurements to ±0.1 ppm for CO2 (±0.05 ppm in the southern hemisphere) and ±2 ppb for CH4 (WMO, 2016) between laboratories. This compatibility between laboratories is ensured if individual laboratories keep uncertainties below half of these goals, which corresponds to the so-called internal reproducibility goals."

Addition to figure captions: "The dashed lines correspond to the WMO internal reproducibility goals, in the case of CO2 in the northern hemisphere (WMO, 2016). Keeping the bias between calibration scale and measurement within these goals ensures achieving the inter-laboratory compatibility goals."

Comment:
Page 4, lines 24-25: You state that the pressure of the external sensor was adjusted to be within a few hPa the same as inside the cavity by a needle valve. Please be more specific. How close was it?

Response:
The values were:
Droplet experiments (Picarro #1, type: flight-ready): -3.5 hPa
Stable levels, Picarro #1: -3.5 hPa
Stable levels, Picarro #2 (type: regular): +18.6 … 20.3 hPa
Stable levels, Picarro #3 (type: flight-ready): -12.5 hPa
The range given for Picarro #2 reflects the drift of the external pressure sensor readings during this experiment, which was larger than during the others.
The main reason for matching cavity pressure closely is that the inlet/outlet (regular/flight-ready analyzer, respectively) valve should not act as a choke, because otherwise the external sensor would not react to cavity pressure changes. The precise pressure difference is not important. Therefore, we will not add the values to the manuscript.

Comment:
The optimal position for an external pressure measurement would be either between the cavity and the inlet or outlet valve, which would allow for the measurement of the same

pressure as in the cavity without the influence of the loop feedback. Would that be feasible, and if yes, why was it not realized?

Response:
In principle, positioning the external pressure sensor between the regulating valve and the cavity would be possible, and we considered this option because of the advantages mentioned by the referee. However, opening tubing connections between these valves and the cavity would risk contamination of the cavity, which would be expensive and time-consuming to fix. This setup may also interfere with the temperature control of the cavity. The cavity and the connectors in question are located inside the so-called "hot box", and the temperature control mechanism of the cavity relies on a stable temperature around 45° inside the hot box. Installing the external pressure sensor between cavity and outlet valve would require extra tubing to leave the hot box, so it may have to be modified to minimize heat exchange with the surrounding. Due to these hurdles, we decided against this option and performed the experiments with the external pressure sensor mounted outside of the Picarro analyzers.

Comment:
Page 5, line 31: What was the reason for the drift of the external pressure sensor? Could this be identified?

Response:
We did not identify the reason for the drift of the external pressure sensor readings. One possible explanation is that the sensor may have reacted to ambient temperature variations, which may have affected the needle valves used as chokes. As stated in the manuscript, $CO_2$ results may have been affected by dissolution in and outgassing from the water reservoir in experiments with stable water vapor levels, which may point to temperature fluctuations as well. However, ambient temperature data are not available and the experiments were conducted in an air-conditioned laboratory. The setup had an influence on the drift; it was significantly larger during the experiment with analyzer #2 (~1.6 hPa), where the external sensor was mounted upstream of the cavity instead of downstream, as it was the case with the flight-ready analyzers #1 and #3 (~0.1 and ~0.2 hPa). Ultimately, we did not answer this question because, as stated in the manuscript, the agreement of results based on the external pressure sensor with those derived from spectroscopic pressure measurements with the oxygen analyzer gives us confidence that the drift of the external sensor readings did not affect our conclusions.

Comment:
Page 18, section 3.4.3: Why is Picarro 4 performing better than others? Is it newer? Is it a different model (according to Table 1 G2401-mc; I could not find any information on a G2401-mc model on the Picarro website, only for G2401-m).

Response:
The "-c" stands for "custom engineering", which refers to modifications of cable harnesses or other details needed to make this instrument suitable for commercial flight. There is no difference between this analyzer and the other flight-ready analyzers that could affect the water correction.
To avoid confusion, we will edit the label to read "G2401-m".

Comment:
Page 15, Table 6: Should the range for CH4 be 0.41 –(-0.86) = 1.27 (instead of 1.30)?.

Response:
This is correct. The range was erroneously rounded to two significant digits. The actual range rounded to three digits was 1.27 ppb.

Comment:
Page 28, line 29: Reference of Stavert et al. is incomplete (journal is missing).

Response:
Thanks, we will fix this.

Comment:
Page 29, line 5: Link to report is wrong.

Response:
We cannot reproduce this error; the link takes us to the correct pdf-document.

References

Andrews, A. E., Kofler, J. D., Trudeau, M. E., Williams, J. C., Neff, D. H., Masarie, K. A., Chao, D. Y., Kitzis, D., Novelli, P. C., Zhao, C. L., Dlugokencky, E. J., Lang, P. M., Crotwell, M. J., Fischer, M. L., Parker, M. J., Lee, J. T., Baumann, D. D., Desai, A. R., Stanier, C. O., De Wekker, S. F. J., Wolfe, D. E., Munger, J. W. and Tans, P. P.: $CO_2$, CO, and $CH_4$ measurements from tall towers in the NOAA earth system research laboratory's global greenhouse gas reference network: Instrumentation, uncertainty analysis, and recommendations for future high-accuracy greenhouse gas, Atmos. Meas. Tech., 7(2), 647–687, doi:10.5194/amt-7-647-2014, 2014.

Stavert, A. R., O'Doherty, S., Stanley, K., Young, D., Manning, A. J., Lunt, M. F., Rennick, C. and Arnold, T.: UK greenhouse gas measurements at two new tall towers for aiding emissions verification, Atmos. Meas. Tech. Discuss., (in review), doi:10.5194/amt-2018-140, 2018.

Verhulst, K. R., Karion, A., Kim, J., Salameh, P. K., Keeling, R. F., Newman, S., Miller, J., Sloop, C., Pongetti, T., Rao, P., Wong, C., Hopkins, F. M., Yadav, V., Weiss, R. F., Duren, R. M. and Miller, C. E.: Carbon dioxide and methane measurements from the Los Angeles Megacity Carbon Project – Part 1: calibration, urban enhancements, and uncertainty estimates, Atmos. Chem. Phys., 17(13), 8313–8341, doi:10.5194/acp-17-8313-2017, 2017.

Welp, L. R., Keeling, R. F., Weiss, R. F., Paplawsky, W. and Heckman, S.: Design and performance of a Nafion dryer for continuous operation at CO2and CH4 air monitoring sites, Atmos. Meas. Tech., 6(5), 1217–1226, doi:10.5194/amt-6-1217-2013, 2013.

Winderlich, J., Chen, H., Gerbig, C., Seifert, T., Kolle, O., Lavrič, J. V., Kaiser, C., Höfer, A. and Heimann, M.: Continuous low-maintenance $CO_2/CH_4/H_2O$ measurements at the Zotino Tall Tower Observatory (ZOTTO) in Central Siberia, Atmos. Meas. Tech., 3(4), 1113–1128, doi:10.5194/amt-3-1113-2010, 2010.

Yver Kwok, C., Laurent, O., Guemri, A., Philippon, C., Wastine, B., Rella, C. W., Vuillemin, C., Truong, F., Delmotte, M., Kazan, V., Darding, M., Lebègue, B., Kaiser, C., Xueref-Remy, I. and Ramonet, M.: Comprehensive laboratory and field testing of cavity ring-down spectroscopy analyzers measuring $H_2O$, $CO_2$, $CH_4$ and CO, Atmos. Meas. Tech., 8(9), 3867–3892, doi:10.5194/amt-8-3867-2015, 2015.

---

## Editor Decision (ED1)

Remaining minor points:

Page 1, line 16: change "of sample" to "between sample"

P2, L2: "This compatibility between laboratories is ensured .." -> "This compatibility is ensured .."

P2, L19: "In many previous" -> "In previous"

P2, end of Introduction section: I suggest to formulate the hypothesis that the biases are due to a sensitivity of the internal cavity pressure to water vapor already at this point. Actually, you should explain that, depending on water vapor, the internal cavity pressure sensor produces an erroneous reading, which translates into a bias in dry CO2 and CH4. Then explain that experiments were designed to show this issue and to characterize the biases, which ultimately allowed you to formulate a correction model. Without this, the paper is hard to read, since many experiments and results start to make sense only later in the text.

P3, caption of Table 1: "in and experiments" -> "in experiments"

P3, L5: The experiments listed in Table 1 should be better motivated, rather than just stating that experiments were conducted with five Picarros. Please explain the purpose of these experiments. Table 1 is very hard to understand without a brief motivation of the individual experiments. Please also explain the meaning of "usable trace gas measurements" (4th column in Table 1). Why would one list experiments that were not "usable" at all?

P3, Table 1: The table suggests that no H2O experiment was conducted with the O2 Picarro #6, in contradiction to the results presented in Sect. 3.2.3.

P4, L10: Shouldn't it be "rather stable"?

P4, L11: Mentioning the fact that CO2 and CH4 readings from this experiment were not used seems irrelevant here.

P4, L20: The setup with external pressure sensor doesn't look very complex to me. Wouldn't it be better to write "Due to issues with this setup explained later, .."

Section 2.3.1: Please explain why the external pressure sensor was placed before the inlet valve (or after the outlet valve) of the cavity (as in the response to the reviewer), since this placement is clearly not optimal. Then explain that this allowed monitoring cavity pressure "indirectly" and that the relation between internal cavity pressure and external pressure sensor was established/calibrated in separate experiments with dry air.

P5, L4-6: I couldn't find any indications on Picarro datasheets that the G2207-i instrument returns information on O2 line width and the optical path length. Does this require operating the instrument in a special mode, or is this part of the housekeeping data?

P5, L13: replace "scale" by "magnitude"

P5, L16: Why do you say "We therefore expect a linear dependence"? Did these studies suggest a linear dependence? If so, please reformulate to make this point clearer.

P5, L26: "their range" -> "the range". This sentence is unclear to me: How can there be a "range" between dry and humid air experiments, if the internal cavity pressure is always regulated to the same value?

Section 2.4, last sentence: Change into a regular sentence (brackets are not needed).

P6, L4: Why should the reader be interested in the median value of 40 min?

P6, L5: Probably one should say "drifted relative to the internal cavity pressure on a timescale ..". What do you mean by "they were calibrated"? Calibrated against what?

P6, L6: It should better be explained why "average readings" were used. Because they were temporally centered on the humidity experiments?

P7, Section 2.6: What was the motivation for using two different $H_2O$ ranges for the cycles?

P8, L8: The title of Section 3.1 is very unspecific.

P8, L10: Probably one should add "as expected" at the end of the sentence.

P8, L13: "were very similar to" -> "differed by only a few percent from"

P8, L14: "this analyzer". Which one?

P9, L6: As mentioned earlier, the paper is confusing if the hypothesis that the internal cavity pressure sensor produces erroneous readings depending on water vapor levels is not formulated earlier on in the manuscript. The sentence starting with "Cavity pressure was estimated .." is a good example for this. You should state again, that since the internal pressure reading was suspected to be wrong, the cavity pressure was additionally estimated based on external pressure readings.

P11, L7: "throughout the experiment". At this stage it is not clear (anymore) what type of experiment this was.

P12, Caption of Fig. 6: Change "The slopes" to "The slopes of the linear parts of the two methods ..".

P13, L15: "yields" -> ", which yields"

P14, caption of Fig. 7: The last line of this caption should be moved to the main text.

P15, Table 5: I don't really understand how the standard deviation for the expanded pressure correction model can be larger (in the case of CH4) than that of the pressure-correction model. Isn't the expanded pressure correction model directly fitted to the CH4 measurements, so that it should minimize the differences from the individual data points? (the same question applies to Table 8).

P15, L24: I can't make any sense out of the statement within brackets.

P17, L2: "Dashes lines: as" -> "Dashed lines as"

P17, Table 7: The coefficient hp of the "joint correction with data from both experiments" is indicated to be (0.16 +/- 0.04) % $H_2O$. The uncertainty range of this coefficient seems too small, since the coefficients of the two experiments separately (0.079 and 0.26) are outside of the range.

P18, L7: "and expanded" -> "and the expanded"

P23, L7-8: This sentence tells the reader at the same time that there were no differences in response between dry and humid air experiments and that, nevertheless, there might be a water-dependent bias. This is very confusing and needs to be briefly explained here (with more details given in the supplement).

P23, L11: What do you mean by "and CH4 data"?? (should probably be deleted)

P23, L18-21: I don't understand how an experiment with dry air can provide useful information on the question, whether cavity pressure may adjust to a new water vapor level on a time scale longer than that of the humid air experiments. This whole paragraph sounds highly speculative to me. Is this really needed?

P23, L29: delete "instead"

P24, L8-10: This sentence rather belongs to the next section 4.3.

P24, L12: " The standard water correction model caused biases" -> "Applying the standard water correction model resulted in biases"

P24, L14: " directly links cavity pressure" -> " directly links cavity pressure estimated from an external pressure sensor"

P24, L22-23: " was based on the parabolic water correction model from the literature and our" -> " combined the parabolic water correction model from the literature with our"

P25, L5: "may help spotting inconsistencies" -> "provides useful information on potential inconsistencies".

P25, L7: You may add that the experiments with stable water vapor levels need to resolve the range of low water vapor levels between 0 to 0.2%.

P25, L12-13: Change sentence " Simultaneously, cavity pressure estimated based on the external pressure sensor was too low and inconsistent in this domain, with the slowest-evaporating droplet closest to the data from experiment with stable water vapor levels" to "Cavity pressure estimated based on the external pressure sensor was lower around the pressure bend position in experiments with fast evaporating droplets than with the slowest-evaporating droplet."
and continue with
"This suggests that the fast water vapor variations .."

P25, L16: "captured exaggerated and inconsistent" -> "tended to exaggerate the"

P25, L18: "slower" -> "more slowly"

P25, L19: delete "than the faster-evaporating droplets"

P25, L24: I suggest to slightly change the structure of Section 4 as follows: Delete title 4.5, change title of 4.5.1 to "4.5 Temporal stability of expanded water correction model" and change title of 4.5.2 to "4.6 Differences of expanded water correction model between analyzers"

P25, L26-29: The sentences referring to the non-useful droplet experiments should be deleted.

P26, L1-2: "far from" -> "well above"

P26, L2-3: delete "between the two experiments" (this should be clear by now)

P26, L9: " with the exception that the effect on CO2 of Picarro #3 appeared diminished" -> "except that the effect on CO2 was reduced for Picarro #3"

P27, L3: "largest at water vapor mole fractions" -> "largest at low water vapor mole fractions"

P27, start of conclusions: I agree with one of the reviewers that it should be stated more clearly that the overall effect is small, especially in the conclusions. Currently, the conclusions section refers to the

WMO compatibility goals (which people assume to be +/- 0.1 ppm (+/-0.05 ppm in S.Hem.) for CO2 and +/-2 2 ppb for CH4), but actually it seems that you are referring to the "internal reproducibility goals", which is only half the compatibility goals.

P27, L12: "reported on here" -> "reported here"

P27, L21: "we used" -> "used"

P27, L22-27: These new sentences are very knotty (therefore, however, therefore) and could probably be reduced to half the length.

---

## Author Response (AR2)

Jena, January 11, 2019

Dear Dr. Brunner,

Thank you for your detailed comments on how to further improve our manuscript. In most cases, we followed your suggestions. Responses to individual comments and questions, as well as reasons where we deviated from your suggestions, are given below in red.

The main text with changes to the previous version of the manuscript highlighted is attached. The supplement was not changed.

With kind regards on behalf of all coauthors,

Friedemann Reum

Remaining minor points:

Page 1, line 16: change "of sample" to "between sample"

Ok.

 P2, L2: "This compatibility between laboratories is ensured .." -> "This compatibility is ensured .."

Ok.

P2, L19: "In many previous" -> "In previous"

Ok.

P2, end of Introduction section: I suggest to formulate the hypothesis that the biases are due to a sensitivity of the internal cavity pressure to water vapor already at this point. Actually, you should explain- that, depending on water vapor, the internal cavity pressure sensor produces an erroneous reading, which translates into a bias in dry CO2 and CH4. Then explain that experiments were designed to show this issue and to characterize the biases, which ultimately allowed you to formulate a correction model. Without this, the paper is hard to read, since many experiments and results start to make sense only later in the text.

We expanded the according paragraph at the end of the introduction.

P3, caption of Table 1: "in and experiments" -> "in experiments"

Our formulation was intended. However, since it was apparently confusing, we changed it to "Overview of experiments performed for this study."

P3, L5: The experiments listed in Table 1 should be better motivated, rather than just stating that experiments were conducted with five Picarros. Please explain the purpose of these experiments. Table 1 is very hard to understand without a brief motivation of the individual experiments. Please also explain the meaning of "usable trace gas measurements" (4th column in Table 1). Why would one list experiments that were not "usable" at all?

Some experiments at an early stage of this work did not yield usable trace gas measurements because they were designed to solely characterize the cavity pressure dependence on water vapor. We added a statement on that to the text of Sect. 2 and added details to the caption of Table 1.

 P3, Table 1: The table suggests that no H2O experiment was conducted with the O2 Picarro #6, in contradiction to the results presented in Sect. 3.2.3.

With Picarro #6, an experiment with stable H2O levels was conducted. The last column of the table refers to this experiment, and we modified its caption to clarify this. We also expanded the entry for Picarro #6 in column 5 to clarify that the "No" refers to the fact that another pressure monitoring method was used, not that no

experiment with varying H2O levels was conducted.

P4, L10: Shouldn't it be "rather stable"?

Since the stability was sufficient we do not think it is necessary to modify "stable" here.

P4, L11: Mentioning the fact that CO2 and CH4 readings from this experiment were not used seems irrelevant here.

We added a motivation why this setup was chosen. The reason was that the experiment was conducted at an early stage of this work and solely served characterizing the cavity pressure dependence on water vapor.

P4, L20: The setup with external pressure sensor doesn't look very complex to me. Wouldn't it be better to write "Due to issues with this setup explained later, .."

We agree that the setup was not too complex. However, its complexity – in particular the way the external pressure measurements were set up – was a major point of concern for one referee of the previously submitted version of this manuscript (https://www.atmos-meas-tech-discuss.net/amt-2017-174/). In our opinion, the referee did not substantiate their diffuse claim that the setup had issues that affected the conclusions, and he/she disregarded our uncertainty analyses that supported our conclusions. Nonetheless, the criticism certainly contributed to the decision not to accept the revised version of this previous manuscript for final publication in AMT. Therefore, we responded to these concerns by developing the spectroscopic methods to measure cavity pressure, eliminating the need for an external pressure sensor altogether. These methods are introduced by the statement in P4, L20-21. Since the results of this second setup confirmed the results of the first, there is no evidence that the first setup had issues that affected the conclusions. Therefore, we think that the current formulation of this statement is appropriate.

Section 2.3.1: Please explain why the external pressure sensor was placed before the inlet valve (or after the outlet valve) of the cavity (as in the response to the reviewer), since this placement is clearly not optimal. Then explain that this allowed monitoring cavity pressure "indirectly" and that the relation between internal cavity pressure and external pressure sensor was established/calibrated in separate experiments with dry air.

We expanded Sect. 2.3.1 accordingly.

P5, L4-6: I couldn't find any indications on Picarro datasheets that the G2207-i instrument returns information on O2 line width and the optical path length. Does this require operating the instrument in a special mode, or is this part of the housekeeping data?

Both quantities are reported by default in standard operating mode along with a variety of other parameters that can be used for diagnostic purposes.

P5, L13: replace "scale" by "magnitude"

Ok.

P5, L16: Why do you say "We therefore expect a linear dependence"? Did these studies suggest a linear dependence? If so, please reformulate to make this point clearer.

Yes, the cited studies inferred linear dependencies of pressure broadening effects on the background gas matrix. We change "to be dependent" to "to be linearly dependent".

P5, L26: "their range" -> "the range".

Ok.

This sentence is unclear to me: How can there be a "range" be- tween dry and humid air experiments, if the internal cavity pressure is always regulated to the same value?

The range was inferred based on the external pressure sensor in first water correction experiments. We added this information as clarification.

Section 2.4, last sentence: Change into a regular sentence (brackets are not needed).

Ok.

P6, L4: Why should the reader be interested in the median value of 40 min?

We think that this information could be helpful for reproducing the experiment.

P6, L5: Probably one should say "drifted relative to the internal cavity pressure on a timescale ..".

Ok.

What do you mean by "they were calibrated"? Calibrated against what?

External pressure sensor readings in humid air were calibrated against external pressure sensor readings in dry air. We added this clarification to the text.

P6, L6: It should better be explained why "average readings" were used. Because they were temporally centered on the humidity experiments?

This was simply a measure for reducing noise (added to the text).

P7, Section 2.6: What was the motivation for using two different H2O ranges for the cycles?

The motivation for the narrow range (0–0.2 % $H_2O$) was to sample the pressure bend at high resolution, and the motivation for the wider range up to 0.8 % $H_2O$ was to sample the transition to a linear dependence of pressure on water vapor. They were not

combined into one measurement because of the drift of the optical phase length measurement.

P8, L8: The title of Section 3.1 is very unspecific.

Changed to "Sensitivities of independent pressure measurements and trace gas readings to changes of internal cavity pressure".

P8, L10: Probably one should add "as expected" at the end of the sentence.

Although this was indeed expected, the expectation was not addressed earlier in the manuscript. Therefore, we think it is clearer not to add "as expected" here.

 P8, L13: "were very similar to" -> "differed by only a few percent from"

Ok.

P8, L14: "this analyzer". Which one?

Changed to "the same analyzer".

P9, L6: As mentioned earlier, the paper is confusing if the hypothesis that the internal cavity pressure sensor produces erroneous readings depending on water vapor levels is not formulated earlier on in the manuscript. The sentence starting with "Cavity pressure was estimated .." is a good example for this. You should state again, that since the internal pressure reading was suspected to be wrong, the cavity pressure was additionally estimated based on external pressure readings.

We added the conclusion that the external pressure sensor revealed biases of the internal pressure sensor to this paragraph:

Before:

"Readings of the internally mounted cavity pressure sensors were, owing to the active pressure stabilization system of the analyzers, stable at 186.65 hPa with standard deviations of 0.02 hPa or less. Cavity pressure was estimated based on external pressure sensor readings and their sensitivity to cavity pressure variations (Sect. 3.1). Cavity pressure estimated in this way varied systematically with the water vapor mole fraction, displaying a uniform pattern for all three analyzers (Fig. 4): …"

Edited version:

"Readings of the internally mounted cavity pressure sensors were, owing to the active pressure stabilization system of the analyzers, stable at 186.65 hPa with standard deviations of 0.02 hPa or less (as expected). However, cavity pressure as estimated based on external pressure sensor readings and their sensitivity to cavity pressure variations (Sect. 3.1) varied systematically with the water vapor mole fraction, revealing that the readings of the internal sensors were biased in the presence of water vapor. Cavity pressure estimated based on the external sensor displayed a uniform pattern for all three analyzers (Fig. 4): … "

P11, L7: "throughout the experiment". At this stage it is not clear (anymore) what type of experiment this was.

Changed to: "In the experiment with the oxygen analyzer (Sect. 2.6), O2 line width measurements obtained for the same humidity levels throughout all cycles were stable (not shown)."

P12, Caption of Fig. 6: Change "The slopes" to "The slopes of the linear parts of the two methods ..".

Changed to "The slopes of the linear parts of the curves… "

P13, L15: "yields" -> ", which yields"

Ok.

 P14, caption of Fig. 7: The last line of this caption should be moved to the main text.

Moved to Sect. 3.4.1.

P15, Table 5: I don't really understand how the standard deviation for the expanded pressure correction model can be larger (in the case of CH4) than that of the pressure-correction model. Isn't the expanded pressure correction model directly fitted to the CH4 measurements, so that it should minimize the differences from the individual data points? (the same question applies to Table 8).

The difference between pressure-correction model and expanded model is that the former makes use of measured cavity pressure (based on the external sensor), while the latter uses the empirical description of these measurements. We think that the slightly better performance when using measured cavity pressure in the case of CH4 indicates small cavity pressure variations during the experiments that were not captured by the empirical description used for the expanded model. However, since the differences are small, they might also be random. Since the differences are irrelevant for the conclusions of the paper, we do not add these speculations to the text.

P15, L24: I can't make any sense out of the statement within brackets.

Changed to: "We also considered using $h_p$ from the 2017 experiment instead, but this induced biases in water-corrected $CH_4$ mole fractions."

P17, L2: "Dashes lines: as" -> "Dashed lines as"

Changed to "The dashed lines are the same as in Fig. 7" in all captions where this was used.

P17, Table 7: The coefficient hp of the "joint correction with data from both experiments" is indicated to be (0.16 +/- 0.04) % H2O. The uncertainty range of this coefficient seems too small, since the coefficients of the two experiments separately (0.079 and 0.26) are outside of the range.

We do not think that the uncertainty estimates are too small. All uncertainty estimates given in Table 7 are standard errors. As explained in the text, $h_p$ could not be determined based on data from the 2015 experiment because not enough data points were obtained to constrain the parameter (this was because the experiment was performed before the cavity pressure hypothesis and the expanded water correction model were developed). Therefore, $h_p$ for the 2015 experiment was taken from cavity pressure data of other experiments (as explained in the text and mentioned in the table) and thus, $h_p$ of the joint water correction should not be compared to $h_p$ given for the 2015 experiment. Regarding the discrepancy between $h_p$ of the 2017 experiment and the latter is within 2 standard errors of the former. Also, the joint correction describes an average of the analyzer responses to water vapor in 2015 and 2017, which might well be different than those of the individual years. Lastly, conclusions are drawn from the variability of the water-corrected data (standard deviations in Table 7), and the coefficients are only given for reference. Therefore, we will not discuss the coefficients of the joint water correction further in the manuscript.

P18, L7: "and expanded" -> "and the expanded"

Ok.

P23, L7-8: This sentence tells the reader at the same time that there were no differences in response between dry and humid air experiments and that, nevertheless, there might be a water-dependent bias. This is very confusing and needs to be briefly explained here (with more details given in the supplement).

The confusion might stem from the term "water-dependent bias", which might not have been explained with enough detail. The sensitivity experiments only tested whether the sensitivities of independent pressure measurements to cavity pressure changes depended on water vapor (which they did not). The term "water-dependent bias" means potential direct sensitivities of the independent pressure measurement methods to water vapor that are unrelated to cavity pressure changes. The experiments with external sensor were designed to prevent such sensitivities, but as explained in Sect. 4.1 and S1, they were not excluded based on direct evidence. In addition, the spectroscopic pressure measurements indeed had linear dependencies on water vapor that were not related to cavity pressure, since the linear parts of the curves $p(h)$ had different slopes across the two methods. However, additional linear dependencies do not affect our conclusions, since they are covered by the water correction models. Only dependencies that would affect the pressure bend would affect the conclusions. The consistency of the pressure bend across multiple methods (external pressure sensor, spectroscopic pressure measurements, CH4 data) gives us confidence that there were no direct sensitivities of the independent pressure measurements on water vapor ("water-dependent biases") that affected our conclusions.

We added these considerations to Sect. 4.1 and replaced the term "water dependent biases" by "direct sensitivities, unrelated to cavity pressure changes, of the independent pressure monitoring methods to water vapor changes". We think that these changes make the section sufficiently clear so no additional changes are made to

the supplement.

P23, L11: What do you mean by "and CH4 data"?? (should probably be deleted)

As explained above, the consistency of the pressure bend as estimated based on external pressure sensor, the spectroscopic methods and CH4 data was a central result that gives us confidence in our conclusions. Therefore, we will leave the current formulation as it is.

P23, L18-21: I don't understand how an experiment with dry air can provide useful information on the question, whether cavity pressure may adjust to a new water vapor level on a time scale longer than that of the humid air experiments. This whole paragraph sounds highly speculative to me. Is this really needed?

We forgot to explicitly point out that the long dry air measurements started after switching from humid to dry air at the end of an experimentation day. Thus, investigating these data provided insight on long equilibration times after switching water vapor levels. We added this information to the text.

P23, L29: delete "instead"

Ok.

P24, L8-10: This sentence rather belongs to the next section 4.3.

The sentence provides closure to the section by explaining the relevance of the results on cavity pressure. We made this a bit more explicit by reformulating the sentence: "Since CO2 and CH4 readings react to changes in cavity pressure, the sensitivity of cavity pressure to water vapor affects CO2 and CH4 readings in humid air. Therefore, the results on cavity pressure imply that an adequate correction method is required to avoid systematic biases in water-corrected dry air mole fractions of CO2 and CH4 due to the cavity pressure dependence on water vapor."

P24, L12: " The standard water correction model caused biases" -> "Applying the standard water correction model resulted in biases"

Ok.

P24, L14: " directly links cavity pressure" -> " directly links cavity pressure estimated from an external pressure sensor"

Changed to "directly links independently estimated cavity pressure…".

P24, L22-23: " was based on the parabolic water correction model from the literature and our" -> " combined the parabolic water correction model from the literature with our"

Ok.

P25, L5: "may help spotting inconsistencies" -> "provides useful information on

potential inconsistencies".

Ok.

P25, L7: You may add that the experiments with stable water vapor levels need to resolve the range of low water vapor levels between 0 to 0.2%.

We added that the range 0 – 0.5 % H2O need to be sampled sufficiently densely at the end of this paragraph.

P25, L12-13: Change sentence " Simultaneously, cavity pressure estimated based on the external pressure sensor was too low and inconsistent in this domain, with the slowest-evaporating droplet closest to the data from experiment with stable water vapor levels" to "Cavity pressure estimated based on the external pressure sensor was lower around the pressure bend position in experiments with fast evaporating droplets than with the slowest-evaporating droplet."

and continue with

 "This suggests that the fast water vapor variations .."

The facts that (1) cavity pressure based on droplets was inconsistent and (2) also the droplet that evaporated comparatively slowly yielded cavity pressures that were lower than those during the experiment with stable water vapor levels are both relevant for the conclusion in the next sentence. We modified the text according to the above suggestion but left both facts in.

P25, L16: "captured exaggerated and inconsistent" -> "tended to exaggerate the"

Similarly as explained in our comment above, the exaggeration was present in all droplet experiments and the inconsistencies are relevant: they substantiate that the droplet experiments yielded unreliable results. Therefore, we will leave both facts in. For clarity, we will change the phrase to "applying the expanded model yielded exaggerated and inconsistent pressure bends".

P25, L18: "slower" -> "more slowly"

Ok.

 P25, L19: delete "than the faster-evaporating droplets"

Ok.

P25, L24: I suggest to slightly change the structure of Section 4 as follows: Delete title 4.5, change title of 4.5.1 to "4.5 Temporal stability of expanded water correction model" and change title of 4.5.2 to "4.6 Differences of expanded water correction model between analyzers"

Ok.

P25, L26-29: The sentences referring to the non-useful droplet experiments should be deleted.

Ok.

P26, L1-2: "far from" -> "well above"

Ok.

P26, L2-3: delete "between the two experiments" (this should be clear by now)

Ok.

P26, L9: " with the exception that the effect on CO2 of Picarro #3 appeared diminished" -> "except that the effect on CO2 was reduced for Picarro #3"

Since we suspect that the apparent reduction of the effect was caused by variations of the CO2 dry air mole fraction delivered to the analyzer (because of the water reservoir), we think the term "appeared" is appropriate here. We will change "diminished" to "reduced" and add a reference to Sect. 4.7, where CO2 results are discussed.

P27, L3: "largest at water vapor mole fractions" -> "largest at low water vapor mole fractions"

Ok.

P27, start of conclusions: I agree with one of the reviewers that it should be stated more clearly that the overall effect is small, especially in the conclusions. Currently, the conclusions section refers to the WMO compatibility goals (which people assume to be +/- 0.1 ppm (+/-0.05 ppm in S.Hem.) for CO2 and +/-2 2 ppb for CH4), but actually it seems that you are referring to the "internal reproducibility goals", which is only half the compatibility goals.

We understand that the magnitude of our correction is small, i.e. in our experiments did not exceed the WMO inter-laboratory compatibility goals. Therefore, we agree that the relevance of the study needs to be stated clearly in the manuscript. The relevance emerges from the fact that there are other errors that affect the accuracy of atmospheric CO2 and CH4 measurements as well and that the WMO goals refer to the combined error. In particular, Yver Kwok et al. (2015) concluded (quote): "Indeed, to be able to reach the WMO comparison goals, we need biases as small as possible for every source of bias". We will edit the conclusions and the abstract to better communicate these arguments, including marking the phrase "as small as possible" as a direct quote from Yver Kwok et al. (2015). The appropriate position for this in the conclusions is the last paragraph, which now reads:

"The biases addressed here are on the order of magnitude of the WMO inter-laboratory compatibility goals. They did not exceed them, but several other error sources that affect GHG measurements, like tracing the calibration of the gas analyzer to a common primary scale (e.g. Andrews et al., 2014), are on the same order of

magnitude. Therefore, to reach the WMO inter-laboratory compatibility goals, biases from each individual error source need to be "as small as possible" (Yver Kwok et al., 2015). Thus, accounting for cavity pressure-related biases of CO2 and CH4 readings contributes to keeping the compatibility of measurements performed with the widely used Picarro GHG analyzers in humid air and potentially in Nafion-dried air within the WMO inter-laboratory compatibility goals."

Similarly, the end of the abstract is now:

"In our experiments, the biases amounted to considerable fractions of the WMO inter-laboratory compatibility goals. Since measurements of dry air mole fractions of CO2 and CH4 are also subject to other uncertainties, correcting the cavity pressure-related biases helps keeping the overall accuracy of measurements obtained with Picarro GHG analyzers in humid and potentially in Nafion-dried air within the WMO goals."

The percentages at the beginning of the conclusions section do refer to the inter-laboratory compatibility goals, not the internal reproducibility goals. To clarify this, we added the specific numbers, i.e. ~0.04 ppm CO2 and ~1 ppb CH4.

P27, L12: "reported on here" -> "reported here"

Since we used "report on" also in the abstract, we prefer the original wording.

P27, L21: "we used" -> "used"

Ok.

P27, L22-27: These new sentences are very knotty (therefore, however, therefore) and could probably be reduced to half the length.

We cut out some details and connected the sentences.

[revised manuscript text omitted]

---

## Author Response (AR3)

Dear Dr. Brunner,

We are delighted to hear that the latest version of our manuscript is suitable for publication in AMT. Below you find our responses to the final small corrections you requested and the updated manuscript with changes to the previous version highlighted. Thank you for your thorough assessments and comments on all versions of our manuscript!

With kind regards on behalf of all coauthors,

Friedemann Reum

Response to editor's comments (in red)

Dear authors,

It is my pleasure to let you know that I consider your manuscript acceptable for publication in its present form. Thank you for the careful replies and amendments. "Gut Ding will Weile haben". Congratulations!

I only have the following tiny technical corrections:
- page 2, line 4: "between" doesn't seem the right word here, probably it should be "in"

To clariy, we changed the sentence as follows:
"This compatibility is ensured if individual laboratories keep errors of measurements with respect to a common calibration scale below half of these goals […]"

- page 4, line 25: change to "solely served to characterize"

Ok.

[revised manuscript text omitted]